**Article** https://doi.org/10.1038/s41467-022-32529-0

# Multivalent interactions between molecular components involved in fast endophilin mediated endocytosis drive protein phase separation

Samsuzzoha Mondal [1], Karthik Narayan [1], Samuel Botterbusch [1], Imania Powers[1], Jason Zheng [1], Honey Priya James[1], Rui Jin[1] & Tobias Baumgart [1]✉

A specific group of transmembrane receptors, including the β1-adrenergic receptor (β1-AR), is internalized through a non-clathrin pathway known as Fast Endophilin Mediated Endocytosis (FEME). A key question is: how does the endocytic machinery assemble and how is it modulated by activated receptors during FEME. Here we show that endophilin, a major regulator of FEME, undergoes a phase transition into liquid-like condensates, which facilitates the formation of multi-protein assemblies by enabling the phase partitioning of endophilin binding proteins. The phase transition can be triggered by specific multivalent binding partners of endophilin in the FEME pathway such as the third intracellular loop (TIL) of the β1-AR, and the C-terminal domain of lamellipodin (LPD). Other endocytic accessory proteins can either partition into, or target interfacial regions of, these condensate droplets, and LPD also phase separates with the actin polymerase VASP. On the membrane, TIL promotes protein clustering in the presence of endophilin and LPD C-terminal domain. Our results demonstrate how the multivalent interactions between endophilin, LPD, and TIL regulate protein assembly formation on the membrane, providing mechanistic insights into the priming and initiation steps of FEME.

Endocytosis is an essential cellular process, such as for maintaining transmembrane receptor homeostasis at the cell surface[1–4]. Endocytic pathways require the local recruitment of adapter proteins at the membrane to form transport carriers, followed by sequestration of the receptors into them[5–7]. Most adapter proteins contain a lipid binding domain for membrane anchoring as well as additional domains to interact with one or more protein binding partners. Functionality of these proteins is regulated by coincidence detection of their binding partners and specific membrane lipids[8–11]. The nature and strength of these molecular interactions play a crucial role in the dynamic assembly and disassembly of endocytic complexes at the plasma membrane.

Clathrin-mediated endocytosis (CME) is the dominant uptake pathway in eukaryotes but various clathrin-independent endocytosis (CIE) pathways exist in parallel that are either cargo-specific or are activated by specific cellular conditions such as receptor hyper-stimulation and stress response[12,13]. Fast endophilin-mediated endocytosis (FEME) is a recently discovered CIE pathway that allows rapid uptake of a subset of G-protein coupled receptors (GPCRs), and receptor tyrosine kinases[13–15]. The β1-adrenergic GPCR (β1-AR) is

[1]Department of Chemistry, University of Pennsylvania, Philadelphia, PA 19104, USA. ✉e-mail: baumgart@sas.upenn.edu

exclusively internalized via FEME[14]. Endophilin, a Bin/Amphiphysin/ Rvs (BAR) domain protein, plays a central role in FEME by driving cargo capture and subsequent generation of membrane curvature. The multifunctionality of endophilin is enabled by its two functional domains—the BAR domain interacts with the membrane and the Src homology 3 (SH3) domain binds to specific target proteins that contain proline-rich-motifs (PRMs)[14,16,17]. FEME requires pre-enrichment of endophilin on the plasma membrane in the form of transient patches formed with the help of an adapter protein called lamellipodin (LPD). Receptor activation further allows endophilin patch interactions with the receptors' PRM-rich third intracellular loop (TIL). At a key step in FEME, TIL binding of endophilin is known to be crucial for receptor internalization[13]. The TIL-endophilin interaction has been suggested to catalyze membrane curvature generation during FEME[13,15]. However, the exact molecular mechanism of how TIL–endophilin interactions facilitate the FEME pathway, is still unclear.

Multivalency is often found to contribute to PRM-containing protein interactions with SH3 domain-containing proteins[18–20]. Endophilin exhibits a bivalency since it exists as a homodimer in solution. This bivalency can be amplified through oligomerization after endophilin binds to the membrane[21,22]. Furthermore, the C-terminal domain of LPD and the TIL region of GPCRs contain multiple PRMs that can interact with endophilin. Multivalent interactions between proteins can lead to liquid–liquid phase separation (LLPS) resulting in a condensed phase coexisting with a dilute aqueous phase[19]. In cells, LLPS drives the formation of membraneless organelles and also has implications in the clustering of signaling molecules on the plasma membrane[23–25]. Actin signaling proteins Nck and N-WASP that contain multiple SH3 and PRMs exhibit a now-classic example for LLPS driven by multivalent interactions[19]. In signaling complexes, LLPS can facilitate sharp transitions in protein functionality by enhancing the local concentration of proteins[19,24–26]. In CME, LLPS has been suggested to serve as a principal mechanism to enhance protein assembly. A recent study by Day et al. reported that LLPS driven by multivalent interactions between FCho1/2 and Eps15 catalyzes the initiation of CME[27]. Here, we asked if similar multivalent interactions between endophilin and its multi-PRM binding partners can lead to LLPS. Specifically, we hypothesized that LLPS promotes the formation of dynamic endophilin-rich clusters on the membrane that could function as priming sites for FEME. The liquid-like clusters could then recruit additional endocytic proteins, such as the activated receptor. Enhanced protein activity at the membrane due to increased local concentrations would then eventually lead to the formation of FEME transport carriers.

In this study, we set out to determine under what conditions endophilin and PRM domain-containing proteins undergo LLPS. We found that endophilin, on its own, forms liquid-like droplets in bulk solution in the presence of molecular crowding agents. The droplets enabled the partitioning of various endophilin binding proteins to be studied. These proteins functioned to either promote or suppress the phase separation. More importantly, endophilin underwent LLPS in bulk solutions even in the absence of crowding agents upon binding to its two major interaction partners in the FEME pathway—(1) the C-terminal intrinsically disordered domain of LPD, represented by a 400 amino acid long region (aa 850–1250)[25] (LPD$^{850-1250}$) that contains several endophilin-SH3 binding sites and (2) the TIL of the β1-AR[14]. We further showed that on a lipid bilayer reconstituted with either TIL or LPD$^{850-1250}$, endophilin formed protein clusters through two-dimensional phase separation. Interestingly, we found that multivalent interactions can regulate endophilin-mediated membrane curvature generation and modulate the size and shape of the generated membrane nanostructures. Altogether, our findings suggest that LLPS mediated by multivalent interactions could be a key mechanism permitting the formation of membrane transport carriers and regulating the membrane curvature generation in FEME.

## Results

### Endophilin alone undergoes LLPS in a crowded environment

We first set out to investigate whether BAR proteins can undergo LLPS by self-interactions under physiological conditions. Over 80% of the proteins in the human proteome are predicted to undergo LLPS either spontaneously or under suitable conditions[28]. The BAR superfamily proteins are well known for their ability to oligomerize on membranes[21,22,29]. There has been no evidence thus far of BAR-proteins undergoing LLPS exclusively through self-interactions. In the cellular environment, macromolecular crowding generates an excluded volume effect that is known to promote LLPS in various proteins by shifting the boundary of phase transition toward lower threshold protein concentrations[30–32]. To test if molecular crowding can promote LLPS in the N-BAR protein endophilin, we introduced the protein (A1 isoform of endophilin, from rat) to a solution containing 10% (w/v) polyethylene glycol (PEG, average molecular weight 8000 Da), a polymeric crowding agent that is commonly used to mimic the cellular crowding environment in vitro[33]. Endophilin was observed to form micron-sized liquid-like droplets at concentrations of 10 μM and higher. The formation of liquid droplets was confirmed by their spherical appearances under transmitted light microscopy, their behavior to undergo coalescence, and their "wetting" behavior on glass surfaces[23] (Fig. 1a and Supplementary Fig. 1a, b). Confocal fluorescence images recorded in the presence of Alexa 594-labeled endophilin (4 mol%) along with unlabeled endophilin (25 μM) showed ~500 times brighter intensities of the droplets compared to the bulk solution (Fig. 1a), demonstrating that the droplets are indeed protein condensates, formed via homotypic (i.e. self-) interactions[34] between endophilin molecules. We constructed a phase diagram by mixing various concentrations of endophilin with different PEG concentrations at room temperature and examining the solution for the presence of liquid-like droplets via transmitted light imaging (Fig. 1b). Lowering PEG concentration from 10% to 2.5% increased the threshold endophilin concentration for LLPS from 10 to 90 μM. The observed PEG dependence of the LLPS boundary is consistent with the notion that molecular crowding promotes phase separation as observed in a variety of different proteins[30–32].

The liquid-like state is typically metastable and protein condensates can further undergo liquid-to-gel-like transitions by establishing long-lived intermolecular interactions[35,36]. That transition is generally facilitated by the presence of crowding agents and results in reduced protein mobility within the droplets[37]. We assessed the translational mobility of Alexa 488-labeled endophilin (1 μM of labeled protein doped with unlabeled protein) within the droplets at various concentrations of PEG by fluorescence recovery after photobleaching (FRAP) measurements (Fig. 1c). The fluorescence recovery profiles were fitted with a double-exponential model for 2.5–7.5% PEG, and with a single-exponential model for 10% PEG (Fig. 1d). The droplets showed partial photorecovery within our time window of monitoring (160 s) suggesting the presence of both fast-diffusing (liquid-like) and slow-diffusing (gel-like) components in the droplets[37,38]. The fraction of the protein in the mobile state (mobile fraction) reduced from 80% to 30% between 5% and 10% PEG suggesting that the liquid-to-gel transition shows a significant PEG concentration dependence within this range (Fig. 1e). Interestingly, the average halftime ($t_{1/2}$) of photorecovery obtained from the exponential fits did not vary significantly with different PEG concentrations, indicating that the diffusion properties of the droplets within our time-scale of observation are dominated by the fast-diffusing component (Supplementary Fig. 1c).

Transitions from liquid to gel-like states in proteins are often associated with misfolding and the formation of amyloid structures[31,32,39]. To test if the gel-like transition in endophilin also occurs via protein conformation changes, we performed circular dichroism (CD) measurements in the presence and absence of 10% PEG (Supplementary Fig. 2a). CD spectra recorded at a protein

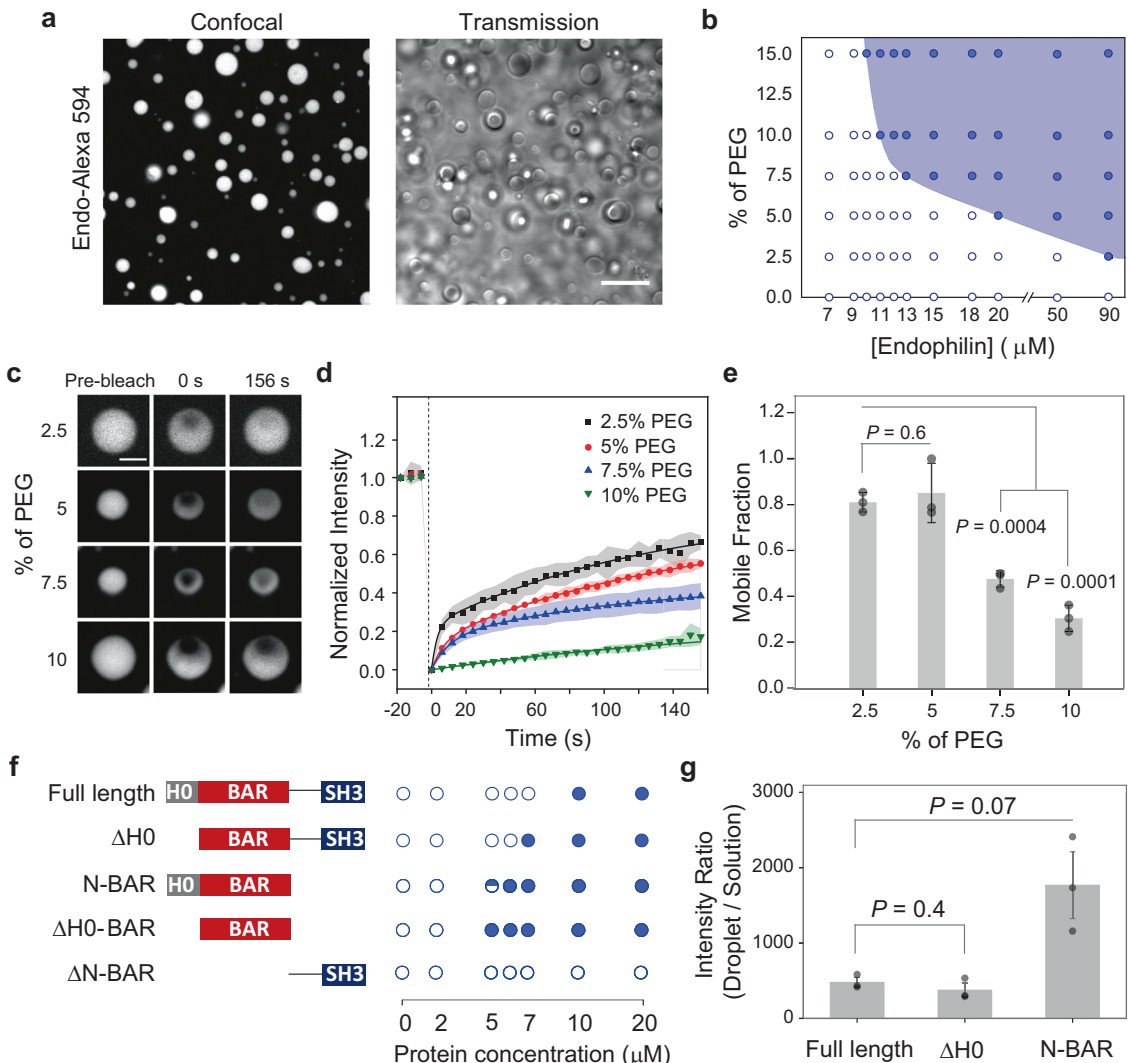

**Fig. 1 | Endophilin undergoes LLPS in a crowded environment. a** Droplets formed by *rat* endophilin A1 (25 μM) in the presence of 10% PEG. Left, confocal fluorescence image of LLPS droplets of endophilin doped with Alexa 594-labeled endophilin (1 μM); right, transmission image of the droplets. Scale bar 10 μm. **b** Phase diagram showing endophilin-PEG LLPS system. The filled circles indicate where liquid-like droplets were observed, whereas the open circles indicate no droplet formation. **c** Representative confocal images of endophilin droplets before, 0 s after, and 156 s after photobleaching at different PEG concentrations. Unlabeled endophilin concentrations used for droplet formation were 150, 50, 25, and 25 μM for 2.5, 5, 7.5, and 10% PEG, respectively, and Alexa 488-labeled endophilin (1 μM) was used for fluorescence imaging. Scale bar 2 μm. **d** FRAP time profiles to show the recovery rate at different PEG concentrations. Normalized intensities of the bleached area relative to the unbleached intensities are plotted for each time point. Each data point is an average of three repeats performed on different droplets, error bars indicate standard deviation. The solid lines indicate 2-exponential (2.5%, 5%, and 7.5% PEG) or 1-exponential (10% PEG) fits of the recovery profiles. **e** Bar plot showing the mobile fractions of protein within the droplets obtained from the exponential fits of the recovery profiles. Error bars indicate standard deviations of three independent FRAP profiles. **f** Domain structures of full-length endophilin, its ΔH0 mutant, N-BAR only mutant, ΔH0-BAR domain mutant, and ΔN-BAR mutant and their phase behavior in the presence of 10% PEG. The open circles and the filled circles indicate observation of no droplet and droplets in 3 out of 3 trials respectively using different preparations whereas the half-filled circle indicates droplets were observed 2 out of 3 trials. **g** Fluorescence intensity ratio of Alexa 594-labeled endophilin variants (1 μM) from corresponding protein droplets vs. from the bulk solution. The droplets were formed in the presence of 25 μM of the unlabeled protein-variant and 10% PEG. The bar plots represent mean ± standard error of mean (s.e.m.) from three independent experiments (gray circles) where 10–20 droplets were considered per experiment. All *P* values (two-tailed) were determined by Student's *t* test, *N* = 3.

concentration of 5 μM, which is below the threshold for LLPS (10 μM), did not show significant differences in the spectral features in the presence or absence of PEG. Above the LLPS threshold (20 μM), the helical features of the spectra remained similar but an overall loss in CD signal intensities at all wavelengths was observed in the presence of PEG, likely caused by a loss in protein concentration in solution due to settling of some droplets (Supplementary Fig. 2b). Overall, the results from our CD experiments suggest that (i) PEG itself does not induce a protein conformation change in endophilin, and (ii) transitioning into the droplet phase does not induce a substantial conformational change for endophilin.

We observed fast gelation kinetics on a minute time scale for endophilin droplets, as indicated by our FRAP studies. About 70% of the protein had matured to a gel-like state within 10 min of mixing the protein with 10% PEG. The timescale of the liquid-to-gel transition is believed to be protein sequence dependent. Under similar conditions (10% PEG), droplets formed by α-synuclein undergo a gel-like transition at much slower rate, over a time course of days, whereas the transition in β-synuclein takes place at a minute time scale like that observed for endophilin[32]. Reducing the PEG concentration to 5% resulted in relatively slower gelation kinetics, and we could observe a significant drop in the mobile fractions between measurements taken

within 10 min and 30 min after mixing. This fraction did not change significantly when monitored over the course of 21 h, indicating that the endophilin droplets reach maturation within an hour of formation (Supplementary Fig. 3a, b). Finally, characterization of the endophilin droplets with two additional alternative techniques, stimulated emission depletion microscopy (STED), and negative stain transmission electron microscopy (TEM) (Supplementary Fig. 3c, d) did not show any inhomogeneous staining or solid-like structures inside the droplets as characterized in the case of disordered proteins such as synuclein or tau[32,39]. Fast transition of endophilin droplets into a gel-like state without showing conformation changes therefore suggests that the protein molecules rapidly reorganize themselves to form strong associations after forming liquid droplets.

## LLPS is driven by the BAR domain itself

With the aim to understand molecular mechanisms behind endophilin self-oligomerization, we asked what functional domains of endophilin are the main drivers for LLPS (Fig. 1f). Cryo-electron microscopy has revealed that endophilin oligomers formed on membrane nanotubes are stabilized by interactions between H0 helices from adjacent N-BAR domains[21]. In addition, the H0 helix is also known to interact with the SH3 domain in solution in the homodimeric form of endophilin[40,41]. We hypothesized that under suitable conditions, H0–H0 and H0–SH3 interactions may lead to crosslinking between the endophilin dimers that can lead to oligomerization. To test whether H0-mediated interactions promote LLPS, we generated a mutant lacking the H0 helix (ΔH0). However, the ΔH0 mutant also formed droplets in the presence of PEG and the threshold concentration of droplet formation was comparable (7 μM) to that of the full-length protein (10 μM) (Fig. 1f). This indicates that the H0 helix has moderate to no contribution to LLPS. We further asked whether LLPS in endophilin is promoted by its 45 amino acid long (aa 248–292) intrinsically disordered linker region that exists between the N-BAR domain and SH3 domain. An endophilin mutant lacking the N-BAR domain (ΔN-BAR, comprising the SH3 domain and its disordered linker) did not form droplets even at the concentration of 70 μM in the presence of 10% PEG (Fig. 1f). Strikingly, an N-BAR-only mutant of endophilin, lacking both the linker and the SH3 domain, formed liquid-like droplets even at a lower threshold concentration (around 5 μM) than the full-length protein (Fig. 1f and Supplementary Fig. 4). Deletion of the H0 region from the N-BAR domain did not cause a significant change to its threshold concentration for phase separation (Fig. 1f). These observations implicated that, rather than interactions between different types of domains of the endophilin protein, it is the self-association behavior specifically of the BAR domain scaffold that drives LLPS in endophilin.

The observed LLPS threshold concentrations for various endophilin mutants mostly remained unchanged between three trials performed using different sample preparations, except for the N-BAR domain that showed a small (order of ~1 μM) variation. The fact that the BAR-domain-only mutants (both N-BAR and ΔH0-BAR) showed LLPS at twofold lower concentration than the full-length endophilin (Fig. 1f) indicated that either the disordered linker or the SH3 domain might suppress droplet formation in the full-length protein. One possibility could be that the disordered linker, via steric effects, interferes with the self-assembly of the N-BAR domains. This notion was further supported by the observation that full-length amphiphysin 1 (Amph1), which contains a much longer (382 aa) disordered linker than endophilin (45 aa), did not form liquid-like droplets in the presence of 10% PEG up to a protein concentration of 60 μM (Supplementary Fig. 5a). To test the hypothesis of an LLPS inhibition effect of BAR protein linkers, we purified an N-BAR-only mutant of Amph1 and introduced it to 10% PEG. Notably, the N-BAR-only mutant of Amph1 formed droplets associated with an LLPS boundary (2 μM) that was comparable with that of endophilin N-BAR (Supplementary Fig. 5b, d). Our hypothesis was further supported by the observation that isoform 9 of

BIN1, another N-BAR family protein that has a linker length comparable to endophilin, underwent LLPS beginning at 5 μM protein concentration in the presence of 10% PEG (Supplementary Fig. 5C). Altogether, these observations strongly indicate that the disordered linker can play an inhibitory role in LLPS driven by the BAR domain. The notion of disordered linkers playing inhibitory roles in BAR-proteins is interesting but not necessarily surprising. While intrinsically disordered regions (IDRs) can favor phase separation in many proteins by allowing conformational flexibilities to form three-dimensional networks they are not always the drivers of phase separation[42,43]. Indeed, it has been shown that IDRs can inhibit LLPS as well[44].

We further compared the relative tendencies of full-length endophilin and the mutants to be in the condensed phase over the dilute phase at a fixed concentration of PEG. With the assumption that Alexa 594 labeled proteins would have similar partitioning abilities as the unlabeled proteins between the condensed and the dilute phases, a minor amount (4 mol%) of labeled proteins was added to unlabeled full-length protein, ΔH0 mutant, and N-BAR-only mutant prior to droplet formation. This enabled estimation of the relative protein densities of the two phases from the ratio of the fluorescence intensities inside and outside droplets. The ratios were similar for full-length endophilin and ΔH0 (480 ± 60 and 380 ± 100, respectively), whereas it was higher (1800 ± 400) for the N-BAR-only mutant (Fig. 1g). A larger density of the N-BAR domain in the condensed phase compared to full-length protein is consistent with the notion that the BAR domain has higher phase separation tendency compared to the full-length protein, as indicated by its phase separation at a lower threshold protein concentration. To test whether any contribution comes from specific interactions between the SH3 domain and other functional domains of the protein, we estimated the tendencies of the SH3 domain itself to partition into the droplets of full-length protein and the two other mutants (Supplementary Fig. 6). Partition coefficients for Alexa 594 labeled SH3 domain were similar and in between 7 and 12 for all three types of droplets, suggesting that the SH3 domain contributes minimally to the partitioning of endophilin into droplets.

## Binding partners of endophilin in FEME pathway partition into LLPS droplets

A protein undergoing LLPS can sequester binding partners as "clients" where the host protein is termed "scaffold"[45]. Relative partitioning tendencies of various clients into a scaffold can be compared via their apparent partition coefficient ($K_{app}$) values[46]. Endophilin interacts with various binding partners in the CME and FEME pathways. The interactions are mainly mediated by the SH3 domain of endophilin, which binds to target PRM-containing proteins[14,47]. While LLPS in endophilin was found to be driven mainly through the N-BAR domain, the SH3 domain can promote sequestration of PRM-containing proteins into the droplets. We hypothesize that phase separation of endophilin facilitates the formation of endocytic protein assembly by allowing partitioning of other endocytic proteins as clients (Fig. 2a).

We first tested the partitioning abilities of two potential clients of endophilin in the FEME pathway. One of these was the 80 amino acid long TIL region of the β1-adrenergic receptor (aa 246–325) which is intrinsically disordered and contains several PxxP motifs (where P is proline, x is any amino acid) which are known to bind endophilin SH3[14,17]. The second client we considered was the C-terminal domain of LPD. The entire C-terminal domain of LPD is a 658 amino acid long (aa 593–1250) disordered sequence that contains 10 PRMs, each 12–13 amino acid long. These PRMs have been shown to bind the endophilin SH3 domain[16]. To facilitate purification, we worked with a relatively shorter (400 aa) fragment of the LPD C-terminal domain (850–1250, LPD[850–1250])[48]. LPD[850–1250] contains 4 out of 10 PRMs and is thus expected to exhibit multivalent interactions with the endophilin-SH3 domain. In order to be able to verify that PRM domains within LPD[850–1250] dominate its interactions with endophilin, we designed a synthetic mimic of the

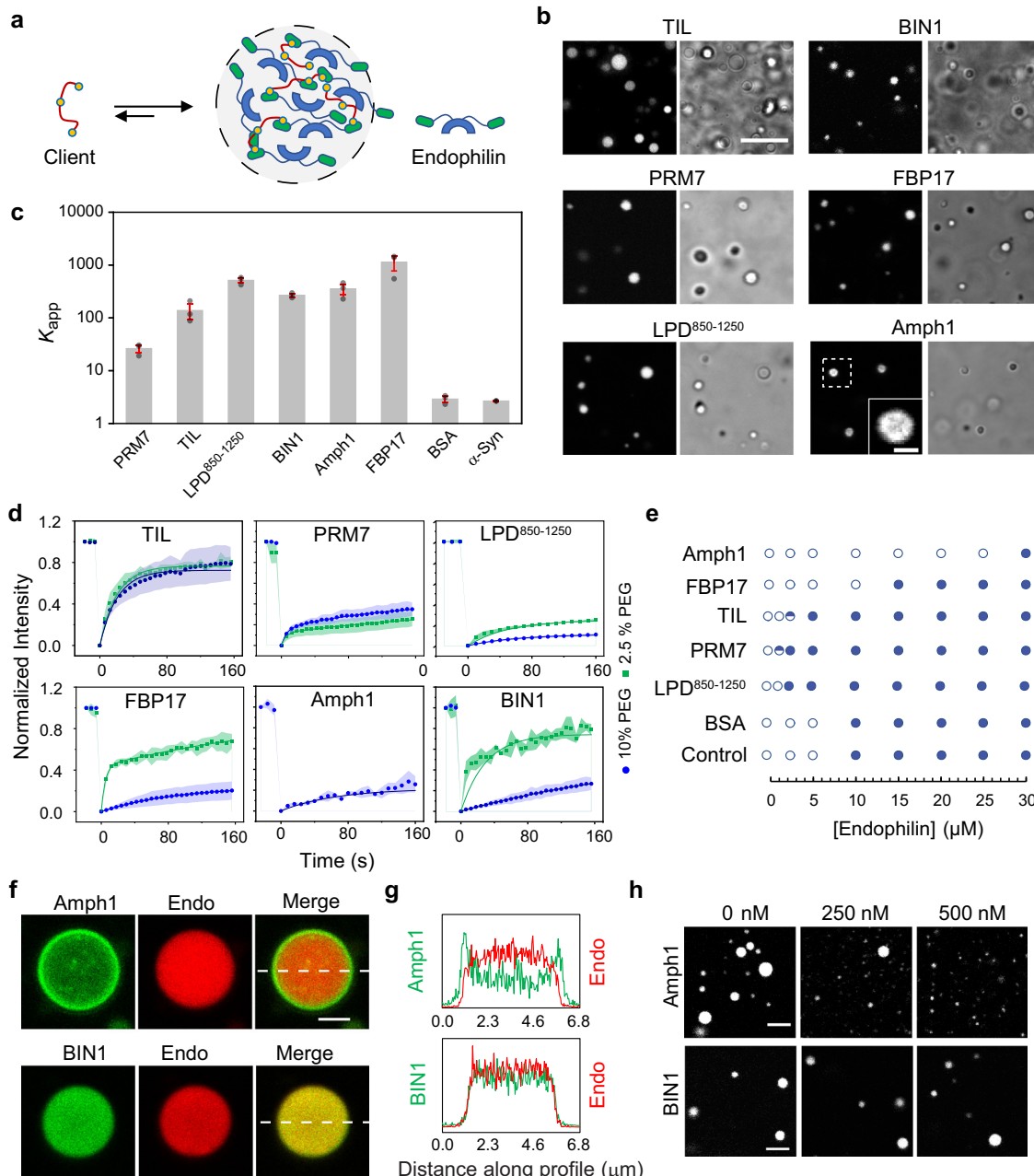

**Fig. 2 | Endophilin binding partners partition into LLPS droplets and exhibit regulatory behavior. a** Graphical illustration of endophilin droplet formed by LLPS allowing partitioning of endophilin binding proteins as clients into the condensed phase. **b** Confocal images showing partitioning of fluorescently labeled TIL (Alexa 488) of β1-adrenergic receptor, PRM7 (Alexa 633) of lamellipodin, LPD$^{850-1250}$ (Alexa 647), BIN1 isoform 9 (Alexa 488), FBP17 (Alexa 488) and amphiphysin (Amph1) (Alexa 488) into droplets formed by endophilin (25 μM) in the presence of 10% PEG. The corresponding transmission images show the endophilin droplets. Scale bar 10 μm. In the case of Amph1, an enlarged image is shown in the inset (scale bar 2 μm) for the droplet surrounded by dotted lines to illustrate the peripheral distribution of the protein. **c** Apparent partition coefficients ($K_{app}$) for the clients within endophilin droplets as determined from fluorescence intensities inside and outside the droplets. Bar plot represents mean ± s.e.m. from three independent experiments (gray circles) where 10–20 droplets were considered per experiment. **d** FRAP profiles of the client proteins partitioned within endophilin droplets formed in the

presence of 2.5% and 10% PEG. Each data point is an average of three repeats performed on different droplets, error bars indicate standard deviation. The solid lines indicate exponential fits of the recovery profiles. **e** Effect of client proteins (10 μM) on the endophilin-PEG phase boundary (10% PEG). The open circles indicate no droplets and the filled circles indicate liquid-like droplets were observed 2 out of 2 independent trials whereas the half-filled circles indicate droplets were observed 1 out of 2 trials. **f** Distribution of Amph1-Alexa 488 (200 nM) and BIN1-Alexa 488 (200 nM) within endophilin droplets (25 μM endophilin, 4% Alexa 594 labeled) formed in the presence of 10% PEG. Scale bar 2 μm. **g** Fluorescence intensity profile along the dotted white line showing Amph1-Alexa 488 fluorescence intensity is higher along the edges of the droplet whereas BIN1-Alexa 488 intensity is homogeneous within the droplet. **h** Confocal images of droplets formed in the presence of 25 μM of endophilin, 10% PEG, and various concentrations of Amph1 and BIN1. Scale bars 5 μm. All experiments were performed in 20 mM HEPES buffer, 150 mM NaCl, 1 mM TCEP, 10% (w/v) PEG, pH 7.4, and at room temperature.

LPD C-terminal domain that consists of a heptameric repeat of a single PRM of LPD (aa 970–981) connected via flexible (Gly-Gly-Ser)$_4$ linkers (PRM7). Fluorescently labeled TIL, LPD$^{850-1250}$, or PRM7 were mixed (4 mol%) with unlabeled endophilin prior to inducing LLPS through PEG (10%). All three peptides showed brighter fluorescence intensities from the droplet phase compared to the bulk phase, indicating they all partition into droplets as clients (Fig. 2b). The $K_{app}$ value for TIL and LPD$^{850-1250}$ (138 ± 40 and 515 ± 60, respectively) indicated that both peptides partition strongly into endophilin droplets, whereas PRM7 showed comparatively weaker ($K_{app}$ 26 ± 4) partitioning (Fig. 2c). Strong partitioning of these binding partners of endophilin into the condensed phase via "scaffold–client" interaction might constitute a key mechanism behind protein sorting in FEME.

In addition to endophilin, various other BAR-domain proteins participate in FEME. Two additional N-BAR family proteins, amphiphysin 1 (Amph1) and BIN1, have been found in endophilin-rich spots both at the leading edge of the plasma membrane and in the majority of FEME carriers formed upon receptor engagement in BSC-1 cells[49,50]. Amph1 was also reported to interact with the SH3 domain of endophilin in vitro via PRMs within its large disordered linker region, and this interaction has been implicated in CME of synaptic vesicles[51,52]. The F-BAR family protein FBP17 is involved in regulating the recruitment of LPD at the leading edge of cells during FEME and is also shown to co-localize with endophilin at sites of CME[49,53]. We then tested the partitioning of Amph1, BIN1 (the ubiquitously expressed isoform 9), or FBP17 into endophilin droplets. All three proteins strongly partitioned into the droplets, with $K_{app}$ values of 352 ± 80, 267 ± 20, and 1130 ± 350, respectively (Fig. 2c and Supplementary Table 1). Interestingly, Amph1 showed an anisotropic partitioning behavior by accumulating preferentially at the droplet periphery compared to the droplet interior (Fig. 2b).

We caution that the ability of partitioning into droplets does not always imply the existence of specific scaffold-client interactions, since protein partitioning into the condensed phase can also be driven by weak, non-specific interactions[45,46]. Accordingly, we further asked, to what extent specific scaffold-client interactions contribute to the client partitioning. To begin to address this, we tested the partitioning of a folded and an intrinsically disordered protein, BSA and α-synuclein respectively, that do not have any specific interaction with endophilin. Both BSA and α-synuclein partitioned into endophilin droplets to a much weaker extent showing $K_{app}$ values of ~3 (Supplementary Table 1), strongly suggesting that protein partitioning into droplets is mainly favored by specific scaffold–client interactions instead (Fig. 2c). These results illustrate the importance of specific protein–protein interactions in the formation of LLPS-driven protein assembly in the endocytic pathway.

We then aimed to explore whether the transition of the scaffold into the gel-like phase affects client mobility within droplets. All clients were allowed to partition into droplets formed in the presence 2.5 and 10% PEG and their FRAP profiles were monitored. The extent of photo-recovery and the PEG concentration dependence on the recovery profile varied between different clients. The full-length protein-based clients, FBP17 and BIN1, showed a stronger PEG concentration dependence in their extent of recovery, whereas among the peptide clients only LPD$^{850-1250}$ showed a moderate difference between 2.5% and 10% PEG (Fig. 2d). A reason why the disordered peptide clients did not show a PEG dependence in their mobility could be that pores of the gel-like network allow greater mobility of the flexible macromolecules than rigid, folded proteins[54] such that the diffusion behavior of the peptides in the gel-like phase is comparable to that in the liquid-like phase. Interestingly, droplets formed at 2.5% PEG in the presence of Amph1 were too small to perform FRAP. However, in the presence of 10% PEG, the droplets were large enough for FRAP experiments and the photorecovery profile of Amph1 was comparable to that of FBP17 and BIN1 at the same % of PEG (Fig. 2d). We discuss the role of Amph1 in

modulating droplet size through interfacial droplet partitioning[55] further below. To summarize, endophilin-binding partners partitioned into the droplets displayed protein-specific diffusion behavior. We next investigated the phase boundary-regulating behavior of these proteins.

## Clients can act as regulators of the LLPS depending on their molecular features

Clients partitioning into the droplet phase can significantly modulate scaffold-scaffold interactions and hence the clients can act as regulators of LLPS[46]. To evaluate this modulation, we determined the threshold endophilin concentration for LLPS (with 10% PEG) in the presence of a fixed client concentration (10 μM) (Fig. 2e). TIL, LPD$^{850-1250}$, and PRM7 were all found to promote endophilin phase separation as the threshold was shifted from 10 to 5 μM or even lower (Fig. 2e). Interestingly, presence of Amph1 increased the threshold concentration for droplet formation 3-fold (from 10 μM to 30 μM) whereas FBP17 had a more moderate effect on the threshold (15 μM). BIN1, at 10 μM concentration, formed droplets in the presence of 10% PEG even in the absence of endophilin. Therefore, the effect of BIN1 on endophilin phase separation could not be compared with the other clients. As expected, the phase boundary remained unchanged in the presence of the weakly partitioning client BSA, reassuring that the regulatory roles exhibited by the clients are dependent on their abilities to form specific interactions with the scaffold.

It is noteworthy that the regulatory behavior of the clients on the LLPS did not show any correlation with their partitioning tendencies ($K_{app}$) into endophilin droplets. Earlier, it was observed that regulators influence LLPS depending on their partitioning abilities into the host protein[46]. Our study further shows that with strongly partitioning clients, molecular features of the client itself and specific interactions between the client and the host proteins could be two major deciding factors in the regulatory action of the client protein. The promotion of LLPS by endophilin in the presence of TIL, LPD$^{850-1250}$, and PRM7 could be attributed to their abilities to form multivalent interactions with the scaffold protein, endophilin, that can promote LLPS by heterotypic interactions in addition to the homotypic interactions[34]. In the case of Amph1, the large, disordered linker that we proposed to inhibit LLPS in Amph1 itself, might either suppress LLPS of endophilin by weakening scaffold–scaffold interactions or reduce the droplet size below the detection limit of transmitted light microscopy.

## Amph1 regulates endophilin droplet size by surfactant-like activity

We further queried how the suppression of LLPS in the presence of Amph1 is related to its unique, anisotropic partitioning behavior into endophilin droplets (Fig. 2c). While Alexa 488 labeled Amph1 showed a higher fluorescence intensity at the droplet periphery compared to the central region, the distribution of endophilin remained homogeneous, as verified from the intensity distribution of Alexa 594 labeled endophilin within the droplets (Fig. 2f, g). The N-BAR domain of Amph1 showed homogeneous partitioning (Supplementary Fig. 8a), indicating that the anisotropic distribution of the protein within endophilin droplets is most likely driven by the disordered linker and the SH3 domain.

FRAP performed on the Amph1 located at the peripheral region of endophilin droplets showed no photorecovery suggesting that Amph1 might form a rigid shell around the droplets (Supplementary Fig. 7). Solid-like shell formation has been reported in multicomponent protein droplets including recently observed intranuclear droplets where RNA-binding protein TDP-43 forms solid shells around a liquid-like core formed by HSP40 family chaperones[56]. Anisotropic distribution of molecular components in multiphase condensates has been attributed to favorable solvent interaction of the shell component over the core components that results in a decrease in surface tension of the overall

system[57,58]. Recently, amphiphilic proteins containing a condensed phase-liking region in addition to a dilute phase-liking region have been shown to form a similar shell-like layer on the condensate surface[55]. Such amphiphilic proteins, having surfactant-like properties, have been shown to regulate the size of biomolecular condensates[55]. Indeed, we observed a significant drop in the number of large droplets (1 μm or above) formed by endophilin (25 μM protein, 10% PEG) in the presence of 0.05–1 μM of Amph1 (Fig. 2h, and Supplementary Fig. 8c). Above 1 μM of Amph1, the total number of endophilin droplets decreased significantly, and no droplets were observed above 2 μM Amph1 (Supplementary Fig. 8b). We asked if a reduction in the droplet size was correlated with a change of the protein volume fractions in the dilute and the condensed phase. To indirectly estimate the protein volume fractions in the dilute and the condensed phases, we separated the droplets formed by centrifugation and estimated the protein concentrations in the dilute phase by Bradford assay. Inhibition of LLPS would cause a reduction in the protein volume fraction in the condensed phase that would be reflected by an increase in the estimated protein concentration in the dilute phase. The estimated protein concentration did not show significant changes in the absence or in the presence of 0.05–1 μM of Amph1 (Supplementary Fig. 8d). These data suggest that the droplet size reductions by amphiphysin (up to 1 μM) can be attributed to a surfactant-like behavior, as opposed to Amph1 inhibiting homotypic LLPS of endophilin.

The observed amphiphilic, surfactant-like properties suggest a potential role of Amph1 as a size regulator of endocytic protein assemblies in FEME. Amph1 is mostly expressed in the brain whereas BIN1 is more ubiquitously expressed and plays an important role in FEME by recruiting Dynein[50]. We have already shown that isoform 9 of BIN1, which has a short linker, similar to endophilin, also undergoes LLPS in the presence of PEG. Unlike Amph1, BIN1 (iso 9) did not show peripheral distribution when partitioned into endophilin droplets (Fig. 2f, g). In addition, we did not observe droplet size regulatory behavior of BIN1 at the concentration range (0.05–1 μM) where Amph1 caused a significant reduction in droplet size (Fig. 2h and Supplementary Fig. 8c). These data strongly suggest that surfactant-like behavior of Amph1 is driven by its long, disordered linker. Interestingly, FBP17 did not exhibit similar surfactant-like behavior. A plausible mechanism would be that due to its longer linker length (100 amino acids longer than FBP17), the change in conformational entropy from the aqueous phase to droplet phase[59] would be more negative for Amph1 than FBP17. Therefore, to minimize the entropic penalty upon droplet partitioning, Amph1 prefers to remain in the interfacial region of a droplet.

**Endophilin undergoes LLPS upon interactions with multiple PRM-containing ligands even in the absence of a crowding agent**

Having observed the inherent abilities of endophilin to undergo LLPS in crowded environments and promotion of the LLPS by its multivalent binding partners, TIL and LPD C-terminal domain, further encouraged us to test whether these two binding partners can drive LLPS in endophilin even in the absence of a crowding environment. We therefore mixed various concentrations of endophilin with equimolar (1:1) concentrations of either TIL or LPD$^{850–1250}$ or PRM7. With TIL we observed the formation of tiny (submicron size) droplets beginning at 20 μM and micron size droplets above 75 μM protein concentration, which implies endophilin TIL interactions drive macroscopic protein-protein phase separation (Fig. 3b). With PRM7, droplet formation began at 10 μM protein concentration. LPD$^{850–1250}$ and endophilin also formed droplets when proteins were mixed at unequal concentrations such as 60 μM of endophilin and 20 μM of LPD$^{850–1250}$ but interestingly, no droplets were observed when these two proteins were mixed at an equimolar ratio at any concentration, including as high as 150 μM. In solution, the droplets coalesced to form larger droplets in a few minutes and upon settling on glass coverslips caused "wetting" within

10 min, indicating their liquid-like behavior. To check protein mobility within the droplets, we performed 2-color FRAP on both endophilin and the peptide components using alternate fluorescent labels. In order to minimize spectral overlap between fluorophores affecting the fluorescence recovery profiles, endophilin labeled with either Alexa 594 (while using TIL-Alex 488; Fig. 3c) or Alexa 488 (while using PRM7-Alexa 633 or LPD$^{850–1250}$-Alexa 647; Fig. 3d, e) were used. TIL, PRM7, and LPD$^{850–1250}$ showed rapid photorecovery and a greater extent of recovery compared to endophilin (Fig. 3c–e). The mobility of endophilin itself in three different types of droplets could not be compared since different fluorescent tags were used to label endophilin in these cases. However, the reduced fractional mobility shown by endophilin in all three types of droplets is indicative of the formation of a gel-like state via BAR domain-driven self-association that we had observed in the case of droplets formed in the presence of PEG (Fig. 1). This implies that the BAR domain driven self-interaction can still take place within droplets formed by multivalent SH3–PRM interactions.

Phase diagrams for endophilin/TIL, endophilin/PRM7, and endophilin/LPD$^{850–1250}$ systems were determined by mixing endophilin and the peptides at various molar ratios and concentrations. Endophilin/TIL and endophilin/PRM7 exhibited a phase boundary with (approximate) reflection symmetry about an axis defined by 1:1 mixing ratio of the proteins/peptides (Fig. 3f–h), consistent with reported phase diagrams of SH3/PRM multimeric system exhibiting heterotypic interactions[46], as well as with theoretical phase behavior predicted for purely heterotypic interactions[34]. The endophilin/LPD$^{850–1250}$ phase boundary, however, was asymmetric with respect to that axis, possibly indicating a competition between homotypic and heterotypic interactions[34]. Interestingly, the endophilin/LPD$^{850–1250}$ system also showed an upper critical concentration above which the droplets disappear (Supplementary Fig. 9a). This kind of closed-loop (reentrant) phase behavior is indicative of heterotypical interactions driving the LLPS process[34]. We mention in passing that the C-terminal domain of LPD shows multivalency not only through its SH3-binding PRMs but that it also contains several FPPPP domains that bind the EVH1 domains of the tetrameric Vasodilator-stimulated phosphoprotein (VASP), an actin regulatory protein found at the cellular leading edge[48]. Indeed, we observed LPD undergo LLPS when mixed with VASP (Supplementary Fig. 9b), indicating that LPD could function as an adapter where multivalent interactions involving phase separation may couple the function of membrane curvature generators and cytoskeletal elements.

We asked whether LLPS in the presence of TIL and LPD is predominantly driven via heterotypic, multivalent interactions with endophilin's SH3 domain or, alternatively, the phase separation is driven by homotypic, endophilin-endophilin interactions, promoted by excluded volume effects caused by these disordered peptides. We determined the volume occupancies of TIL, PRM7, LPD$^{850–1250}$, and PEG from their specific volumes (see Methods) at the threshold concentrations of those molecules required for driving LLPS while using the common reference concentration for endophilin of 20 μM. The estimated volume occupancies of TIL and PRM7 at the corresponding threshold concentrations (20 and 10 μM, respectively, according to Fig. 3e, f) were found to be about 300 times lower than that of PEG (at 5% w/v, see Fig. 1b). Similarly, the estimated volume fraction for LPD$^{850–1250}$ for its threshold concentration to drive LLPS (5 μM) (Fig. 3h) was about 150 times lower than that of 5% PEG. This suggests that the excluded volume effect would have minimal contributions at the concentrations of TIL, PRM7, or LPD$^{850–1250}$ that drive LLPS.

**Endophilin causes clustering of TIL and LPD C-terminal domain on the membrane by two-dimensional phase separation**

Prior to receptor activation, endophilin is recruited by LPD at the leading edge of cells and forms transient clusters that act as FEME priming sites[13]. Following ligand activation, receptors are sorted into

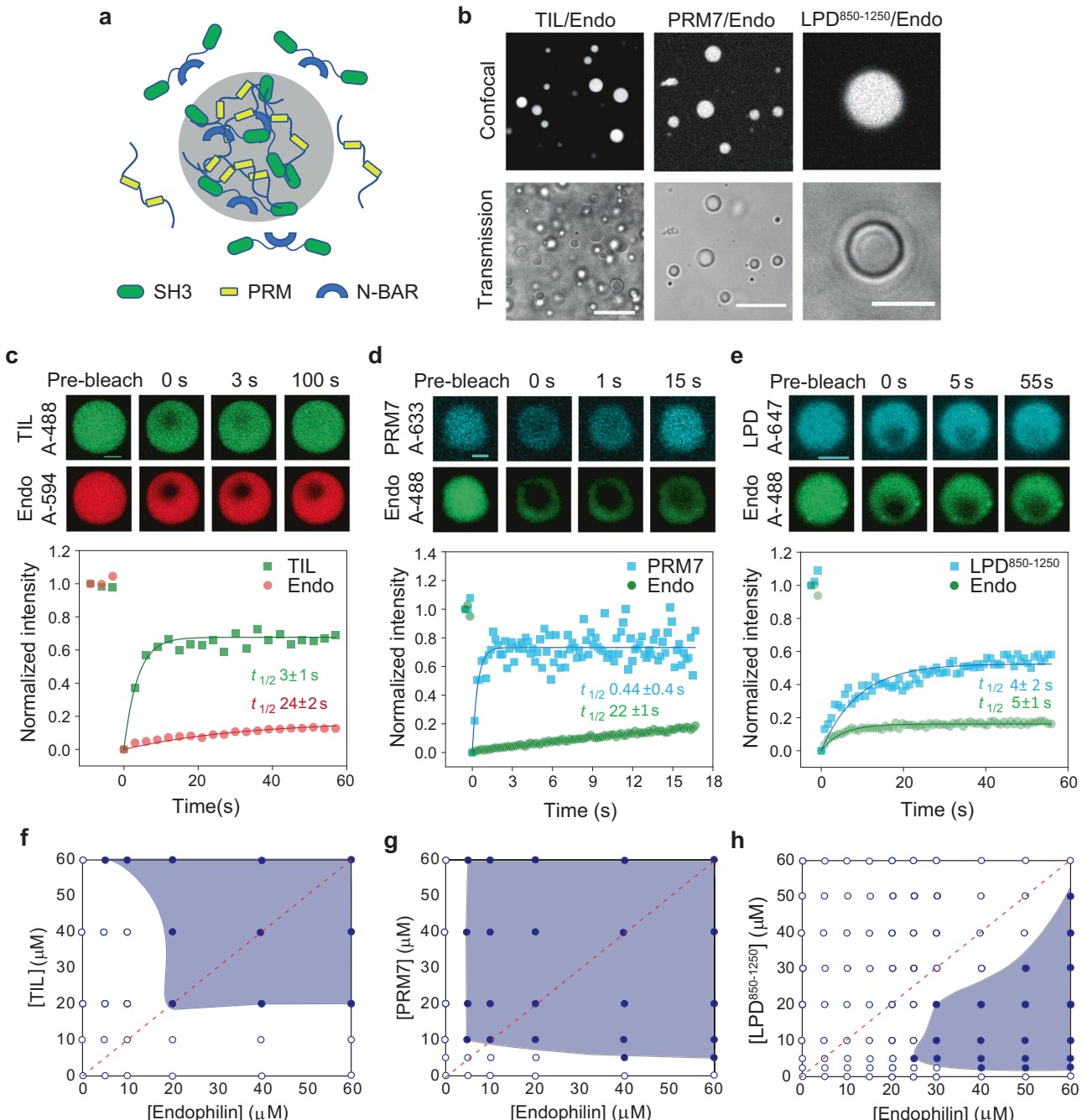

**Fig. 3 | Endophilin undergoes LLPS through multivalent interactions with proline-rich motifs in the absence of PEG. a** Cartoon diagram illustrating that multivalent interaction between SH3 domains of dimeric endophilin and a multiple PRM containing ligand can drive LLPS. **b** Confocal fluorescence images (top) and transmission images (bottom) of droplets formed by the TIL/endophilin, PRM7/ endophilin, and LPD850-1250/endophilin system. The TIL/endophilin and PRM7/ endophilin droplets were formed in the presence of 100 μM of endophilin and 100 μM of either TIL or PRM7 with 1 μM of either TIL-Alexa 488 or PRM7-Alexa 633. The LPD850-1250/endophilin droplets were formed by mixing 20 μM of LPD850-1250 and 60 μM of endophilin and contained 1 μM of LPD850-1250-Alexa 647. Scale bars 20 μm. **c** Confocal images and intensity profiles from a representative FRAP experiment on a TIL–endophilin droplet to monitor the mobility of both endophilin (Alexa 594)

and TIL (Alexa 488). Recovery time constants ($t_{1/2}$) for TIL (green) and endophilin (red) were determined from single-exponential fits (solid lines) of the FRAP data and have been reported as mean±s.d. of three independent FRAP experiments. **d**, **e** FRAP studies on PRM7/endophilin and LPD850-1250/endophilin droplets in the presence of endophilin-Alexa 488, PRM7-Alexa 633, and LPD850-1250-Alexa 647. Recovery time constants ($t_{1/2}$) for PRM7, LPD850-1250 (cyan), and endophilin (green) have been reported as mean±s.d. of three independent FRAP experiments. Scale bars 2 μm (**c**–**e**). **f**–**h** Phase diagrams for TIL/endophilin (**f**), PRM7/endophilin (**g**), and the LPD850-1250/endophilin (**h**) LLPS systems. The red dotted line represents the axis of a 1:1 mixing ratio of both proteins. All experiments were performed in 20 mM HEPES buffer, 150 mM NaCl, 1 mM TCEP, pH 7.4, and at room temperature.

endophilin-rich clusters through TIL-SH3 interactions (Fig. 4a). From our bulk experiments, it is evident that endophilin forms liquid-like droplets upon multivalent interactions with both LPD C-terminal domain and TIL. We hypothesized that on the membrane, such

multivalent interactions play a crucial role in the formation of endophilin-rich clusters with LPD and during the sorting of receptors into the clusters. Earlier studies have shown that certain signaling proteins can undergo LLPS via multivalent interactions to form

submicron-sized clusters on the membrane[20,24]. We first determined to what extent endophilin can cause clustering of TIL, LPD[850–1250], or PRM7) on the membrane. For this study, we prepared solid supported bilayers (SSBs) with tethered TIL-His$_6$, PRM7-His$_6$, or LPD[850–1250]-His$_6$ via Ni[2+]-NTA lipids (1 mol% of total lipids) (Fig. 4b). Under our optimized coupling conditions, TIL density on the bilayer was found to be $330 \pm 15$ molecules per $\mu m^2$ that led to a membrane area coverage of 0.8% (see "Methods"). All three peptides were uniformly distributed and freely mobile on the bilayers (Fig. 4c and Supplementary Fig. 10a). FRAP studies showed TIL and LPD[850–1250] both exhibited a comparatively faster ($t_{1/2}$ ~ 7 s), and greater extent of photo-recovery (92–94%) than PRM7 ($t_{1/2}$ 18 s, 56% recovery). The slower mobility of PRM7 on the bilayer might be attributed to its specific amino acid sequence, or possibly to stronger membrane attraction of the conjugated fluorophore (Alexa 633) compared to the fluorescent labels on TIL (Alexa 488) and LPD[850–1250] (Alexa 647) as reported earlier[60].

To study the effect of endophilin on the bilayers, we utilized the ΔH0 mutant of endophilin instead of the full-length protein since the latter was shown to destroy supported lipid bilayers by insertion of its H0 helix[61]. In bulk experiments, the ΔH0 mutant of endophilin showed similar phase behavior as the full-length protein and also formed LLPS droplets in the presence of TIL (Fig. 1f and Supplementary Fig. 11). Within 5 min of adding endophilin (ΔH0) we observed submicron sized clusters forming in all three types of bilayers. When imaged after 30 min, the clusters looked bigger and brighter and the intensity around the clusters depleted, indicating active sequestration of proteins into the clusters from the surroundings (Fig. 4c). Endophilin colocalized with TIL, LPD[850–1250], or PRM7 in the clusters (Fig. 4d, e). The extent of clustering was quantified by a radially averaged autocorrelation function and exponential fits of the correlation function allowed estimation of the correlation length ($R$)[62] (Fig. 4f). A value of $R$ greater than a pixel width (0.082 μm or 82 nm) indicated significant clustering of proteins on the bilayer (Supplementary Table 2). Bilayers conjugated with TIL, LPD[850–1250], and PRM7 showed $R$ values between 30 and 40 nm range in the absence of endophilin, which increased to 90–120 nm in the presence of endophilin (1 μM). Increasing the endophilin concentration from 1 to 2.5 μM caused a moderate increase in $R$ (140–180 nm) (Fig. 4c, f).

The membrane clusters formed well below the phase boundary of endophilin/TIL and endophilin/LPD systems (Fig. 3f–h) in the bulk. Unlike the three-dimensional, spherically shaped bulk condensates (Figs. 1a and 3b), clusters on the membrane were irregularly shaped and smaller in size. Such appearance of the clusters is indicative of the nucleation regime of phase separation on two-dimensional surfaces[24]. The clusters resemble membrane condensates earlier observed with signaling protein complexes[18] and postsynaptic density proteins[63]. The clusters showed partial photobleaching recovery of TIL, PRM7, and LPD[850–1250] (Supplementary Fig. 10b), indicating that the two-dimensional membrane clusters exhibited partially liquid-like and partially gel-like behavior, which was also observed in the case of endophilin driven condensates in the bulk (Figs. 1c–e and 3c–e). Altogether, these results provide strong evidence for our hypothesis that clustering of endophilin in the presence of LPD at the FEME priming sites is driven by phase separation via multivalent interactions between endophilin SH3 and LPD's C-terminal domain. The fact that we observe similar behavior comparing the simple PRM7 peptide (multiple repeats of a single PRM separated by oligo-GGS spacers) and the more complex LPD[850–1250] suggests that the behavior of the latter is dominated by its PRMs.

Moreover, the observation that interactions between endophilin and TIL lead to cluster formation on the membrane suggests TIL might have a synergistic effect on the maturation of transient endophilin clusters into stable transport carriers.

## TIL partitions into endophilin–LPD clusters and enhances protein clustering on the membrane

We have demonstrated that endophilin can drive cluster formation of membrane-coupled TIL and LPD C-terminal domain (Fig. 4). Next, we asked whether endophilin induces co-clustering of membrane-bound TIL and LPD or, alternatively, if these two proteins form separate clusters with endophilin. Surprisingly, TIL showed a tendency to form clusters when introduced to the bilayer along with either LPD[850–1250] or PRM7 even in the absence of endophilin (Supplementary Fig. 12). Clustering could be minimized on a bilayer containing TIL and PRM7 by reducing the solution concentration of TIL (to 50 nM) used for membrane-tethering (Fig. 5a, top panel). However, for bilayers containing TIL and LPD[850–1250] clusters appeared even after lowering the solution concentrations of both TIL and LPD[850–1250] to 50 nM. We verified that the clustering was not caused by the TIL density being too high. Our estimation showed TIL density of $140 \pm 20$ molecules per $\mu m^2$ on the bilayer (in the presence of 50 nM solution concentration), which is comparable to the density of β-adrenergic receptors in cells[64]. TIL alone did not cause clustering when present at 2.4 times higher density (Fig. 4b). Future research would have to investigate if these molecular interactions between TIL and the LPD C-terminal domain are relevant for cellular function.

Bilayers having TIL and PRM7 were used to study whether endophilin causes co-clustering. We confirmed the fluidity of both TIL and PRM7 on the bilayers by FRAP (Supplementary Fig. 13a, b). Within 1 min of incubation with endophilin (ΔH0), cluster formation was observed in fluorescence imaging channels corresponding to TIL, PRM7, and endophilin. Clusters appeared bigger in size and brighter when imaged 15 min after endophilin addition (Fig. 5a, b). TIL showed a comparatively greater extent of clustering ($R = 153$ nm) than PRM7 ($R = 69$ nm) (Fig. 5a–c and Supplementary Table 3). This difference could be due to the stronger membrane affinity of Alexa 633-labeled PRM7 than Alexa 488-labeled TIL on supported bilayer[60] that would slow down its assembly into clusters. Cross-correlation analysis between TIL and PRM7 channels showed significant enhancement in the correlation function after the addition of endophilin (Supplementary Fig. 13C). This data indicates that co-clustering of TIL and PRM7 was indeed promoted by endophilin and not caused by the self-clustering of TIL and PRM7 that was observed at higher concentration of TIL. Co-clustering of TIL and PRM7 indicates that they can act synergistically in cluster formation during the formation of FEME transport carriers. This motivated us to investigate further if TIL can act as an agonist of protein cluster formation at the FEME priming sites that are pre-enriched with endophilin and LPD.

First, we formed supported lipid bilayers containing either tethered PRM7 or tethered LPD[850–1250] and then introduced endophilin to create reconstituted models of FEME priming sites. The introduction of endophilin caused cluster formation on both types of bilayers as expected (Fig. 5d–i; upper panels). Next, TIL (50 nM) was added to the system. Within 5 min of addition, TIL not only accumulated into the pre-existing clusters but also started forming new clusters that were enriched in either LPD[850–1250] or PRM7 along with TIL and endophilin. The clusters appeared bigger and brighter when imaged 15 min after adding TIL, indicating the accumulation of proteins from the membrane surroundings into the clusters (Fig. 5d–i; lower panels). TIL colocalized with both endophilin and LPD C-terminal domain (both LPD[850–1250] and PRM7) in the clusters (Fig. 5e, h; lower panels). Auto-correlation analysis showed ~2 times increase in the $R$ values after the addition of TIL for both endophilin/LPD[850–1250] and endophilin/PRM7 systems (Supplementary Table 4), indicating that TIL causes enhancement in clustering on a membrane pre-enriched with endophilin and LPD (Fig. 5f, i). Cross-correlation analysis illustrated a stronger correlation between endophilin and LPD[850–1250]/PRM7 indicating TIL enhances the co-clustering of LPD and endophilin (Fig. 5j, k; left panels). In addition, strong cross-correlation of TIL with both

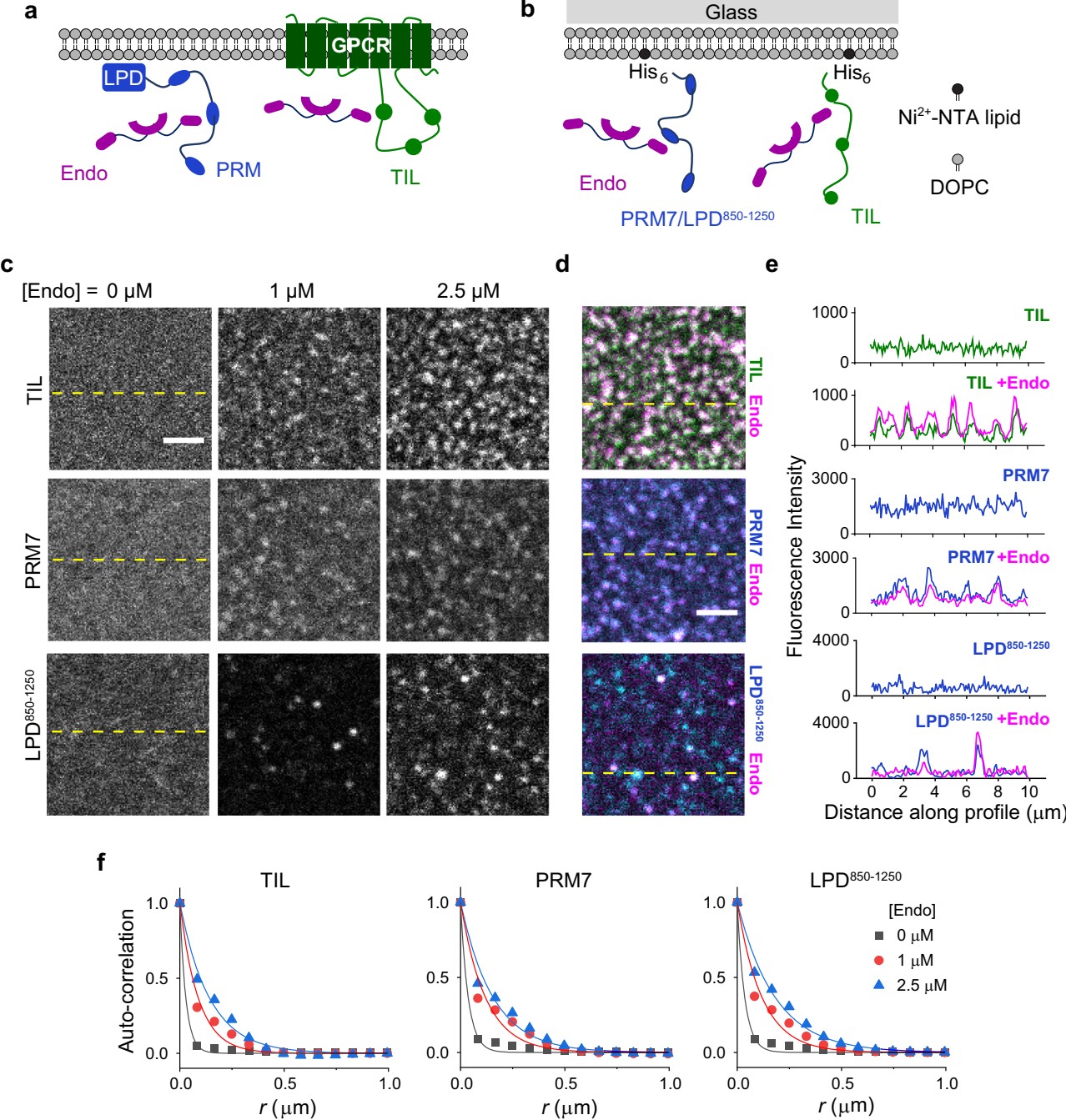

**Fig. 4 | Endophilin forms clusters on the membrane in the presence of multivalent binding partners. a** Cartoon representation of the interactions between endophilin and proline-rich-motifs of GPCR-TIL and LPD C-terminal domain on cell membrane. **b** An in vitro model system that has been developed to mimic the endophilin–LPD–GPCR interactions using solid-supported bilayers (SSBs) with conjugated either PRM7-His$_6$ or LPD$^{850-1250}$-His$_6$ and TIL-His$_6$ via Ni$^{2+}$-NTA-lipids (right). **c** Confocal images showing distributions of TIL-Alexa 488 (top), PRM7-Alexa 633 (middle), and LPD$^{850-1250}$-Alexa 647 (bottom) on SSBs composed of Ni$^{2+}$-NTA lipid and DOPC (1:99). Images were recorded after incubating the functionalized SSBs with 0, 1, and 2.5 μM endophilin for 30 min. Scale bar 2.5 μm. **d** Merged images of endophilin-Alexa 594 channel with TIL-Alexa 488 channel (top), PRM7-Alexa 633 channel (middle), and LPD$^{850-1250}$-Alexa 647 channel (bottom) in the presence of 2.5 μM endophilin. Scale bar 2.5 μm. **e** Intensity profiles along the dashed yellow lines shown in **c**, **d** showing that clustering of TIL, PRM7, and LPD$^{850-1250}$ occurred in the presence of endophilin and endophilin itself colocalized with TIL, PRM7, and LPD$^{850-1250}$ in the clusters. **f** Radially averaged normalized autocorrelation function ($G(r)$) demonstrating the degrees of clustering in the TIL (left), PRM7 (middle), and LPD$^{850-1250}$ (right) channels at 0–2.5 μM of endophilin. The auto-correlation function determines the probability of finding a fluorescent pixel at a given distance $r$ from a center pixel. Solid lines represent the fitting of the auto-correlation plots to a single-exponential function, $G(r) = A\,e^{-r/R}$, to express the extent of clustering in terms of a correlation length ($R$). All experiments were performed in 20 mM HEPES buffer, 150 mM NaCl, 1 mM TCEP, pH 7.4, and at room temperature.

LPD$^{850-1250}$ and PRM7 indicated co-partitioning of TIL and LPD (Fig. 5j, k; right panels). These results altogether imply that TIL can act as an agonist for endophilin-rich clusters on the membrane. Post receptor activation, interactions between receptor TIL with endophilin pre-

enriched at the priming sites have been proposed as an essential step for the initiation of FEME[13,14]. Based on the observations of our minimalist in vitro reconstitution model, we propose here that the enhancement of clustering upon engagement of the TIL of β1-AR into

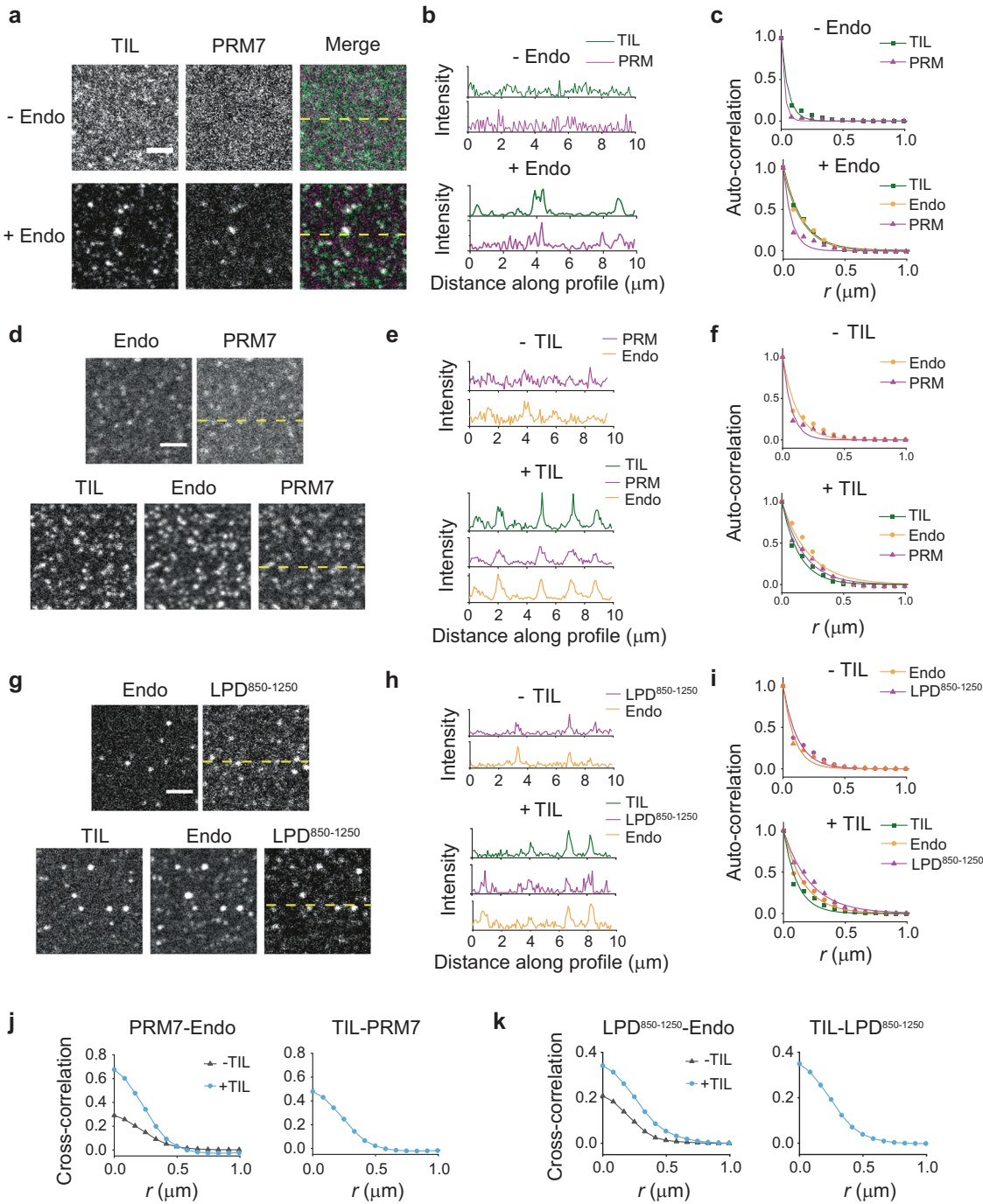

**Fig. 5 | TIL partitions into pre-existing endophilin–LPD clusters on the membrane and further enhances clustering. a** Confocal images of SSBs with conjugated His$_6$-tagged TIL (Alexa 488) and PRM7 (Alexa 633) in the presence and absence of endophilin (10% Alexa 594 labeled). Scale bar 2.5 μm. **b** Fluorescence intensity profiles for the images along the dashed yellow lines shown in **a** demonstrating the extent of co-localization of TIL and PRM7 within the clusters. **c** Radially averaged normalized autocorrelation functions and its single exponential fits (solid lines) demonstrating the clustering in the TIL, PRM7, and endophilin channels before (left) and after (right) addition of endophilin. **d** Protein distribution on the SSBs containing tethered PRM7 and endophilin before and 15 min after addition of TIL (50 nM). **e** Fluorescence intensity profiles for the images along the dashed yellow lines shown in **e** showing co-localization of TIL and PRM7 into the clusters. **f** Extent of clustering in the TIL, PRM7 and endophilin channels before (top) and after (bottom) addition of TIL quantified by radially averaged

autocorrelation function and its single-exponential fits. **g** Distribution of tethered LPD$^{850-1250}$ and endophilin on SSBs before (top) and after (bottom) addition of TIL (50 nM). **h** Fluorescence intensity profiles along the yellow dashed lines shown in **g**. **i** Radially averaged auto-correlation functions with fits to show the extent of clustering in endophilin, LPD$^{850-1250}$, and TIL channels before (top) and after (bottom) addition of TIL. **j** Left, cross-correlation functions to compare the extent of co-clustering between pre-existing PRM7 and endophilin on a bilayer before (gray) and after (cyan) addition of TIL. The extent of co-clustering between the added TIL with the pre-existing PRM7 is shown on the right. Solid lines are the guide for the eye. **k** Cross-correlation analysis for the LPD$^{850-1250}$/endophilin on a bilayer before (gray) and after (cyan) addition of TIL. On the right, the extent of co-clustering between the added TIL and the pre-existing LPD$^{850-1250}$ is shown. All experiments were performed in 20 mM HEPES buffer, 150 mM NaCl, 1 mM TCEP, pH 7.4, and at room temperature.

endophilin and LPD assembly at the priming sites could be the key to the transition from priming to the carrier formation stage in FEME.

## Endophilin interacts with LPD on membranes to support membrane-membrane adhesion and budding necks

Endophilin, like other BAR-family proteins, generates and stabilizes membrane curvature with its crescent-shaped BAR-domain dimer[41,65,66]. It spontaneously generates micron-long tubules of narrow radius (~30 nm) in vitro[65]. However, endophilin activity during FEME is regulated by its binding partners such as LPD, and the TIL of specific receptors that are internalized via this pathway[13–15]. It is largely unclear what molecular mechanism regulates the curvature generation and sensing properties of endophilin in cells. We formulated three hypotheses for how endophilin and lamellipodin might interact on (and with) membranes and tested them in turn: (1) LPD enhances endophilin's curvature generation capacity through local enrichment, (2) phase separation of LPD generates negative membrane mean curvature (leading to interior tubules, as has been observed for several membrane-bound proteins known to phase separate[67]) and (3) multivalent interactions between membrane-bound endophilin and endophilin-bound lamellipodin can stabilize membrane–membrane adhesion and negative Gaussian curvature (i.e., local saddle shapes) at the necks of budding endocytic vesicles and tubules.

To test the first hypothesis, we compared the membrane tubulation properties of endophilin in the presence and absence of LPD[850–1250] on giant unilamellar vesicles (GUVs). GUVs composed of anionic phospholipid DOPS and zwitterionic lipids DOPE and DOPC in a molar ratio of 45:30:25 formed micron-length tubules when incubated with endophilin (Fig. 6a, b). The addition of LPD[850–1250] to the GUVs resulted in the recruitment of LPD both onto the GUV surface and the tubules (Fig. 6a, c). Notably, the binding of LPD caused apparent shortening of tubule length and the long tubules disappeared leaving a few clusters on the GUV surface (Fig. 6b and Supplementary Movie 1). A simultaneous enhancement in the LPD (Alexa 647) intensity on the GUV membrane indicated that the tubule contraction is indeed LPD binding mediated (Fig. 6c). Fluorescence intensity of endophilin (Alexa 488) on the membrane remained constant indicating that LPD[850–1250] present in the solution did not induce unbinding of endophilin from the membrane resulting into apparent disappearance of the tubules (Supplementary Fig. 15a). These observations lead us to exclude hypothesis (1) from above: LPD-endophilin interactions do not seem to enhance the membrane tubule (positive mean curvature) generation ability of endophilin. This conclusion is consistent with an earlier hypothesis regarding transient LPD-endophilin FEME priming sites as locally flat patches[15]. The image sequence in Fig. 6a also allows to test our hypothesis (2) from above. Several intrinsically disordered proteins undergoing phase separation on GUV membranes have been shown to cause inward membrane bending (negative mean curvature) and inner tubulation[67]. We found no evidence that similar behavior is displayed by LPD[850–1250]: no interior tubulation was generated (see Fig. 6a and Supplementary Fig. 14). Therefore, LPD-mediated phase separation does not seem to generate negative membrane mean curvature for the conditions we have explored. Whether these conclusions hold true throughout the compositional phase space will have to be addressed in future studies. Finally, we discuss our hypothesis 3) from above, which is motivated by the following findings.

After the addition of LPD to endophilin-coated vesicles, we observed two GUVs that were connected via a long membrane tether to pull each other closer together, suggesting an increased membrane tension induced by the endophilin–LPD interaction (Supplementary Movie 2). This is consistent with the shortening of endophilin-generated tubules described above. How might such a membrane tension be induced by endophilin–LPD interactions? To answer that question, we performed TEM imaging of LUVs that are tabulated in the presence of endophilin and compared them in the presence and absence of LPD[850–1250]. In the absence of LPD, separate tubules of micron-length were observed as expected (Fig. 6d). In the presence of LPD[850–1250], we observed clusters of multiple LUVs adhered together (Fig. 6e, left panel, and Supplementary Fig. 15b). Along with LUV clusters, we also observed tubules adhered along their length (Fig. 6e, right panel). These observations suggest that LPD[850–1250] causes adherence of membranes coated with endophilin, most likely via forming multivalent interactions between endophilin molecules present on opposing bilayers. If a multitude of tubules and buds exist on a GUV, adhesion-induced wrapping[68] of such structures via the membrane of a vesicle with fixed volume would increase membrane tension as observed above.

Membrane adhesion could support the formation of membrane buds at FEME sites[69] (Fig. 6f). The junction of two LUVs observed in our TEM images resembles the neck region of such membrane buds where negative Gaussian membrane curvature is generated. While endophilin by itself typically stabilizes positive membrane curvature, the multivalent LPD could enable endophilin to support the neck area with negative Gaussian membrane curvature, by facilitating the local adhesion of the opposing membrane sections in the neck region (Fig. 6f). Complex and competing mechanical interactions, including membrane tension, bending resistance, and cytoskeletal forces, likely determine the fate of a budding site[6,15]. This interplay will be a target for future studies.

## Discussion

The importance of LLPS is evident from its association with an increasing number of biological phenomena ranging from the formation of membraneless cytosolic organelles to the clustering of signaling proteins on the membrane[35,45]. LLPS is facilitated by multivalent interactions that can be achieved through either self-association or binding with other multivalent ligands[19]. Recently, LLPS has been considered to serve as a key mechanism of protein assembly formation in CME[27]. Here we demonstrate that the BAR-protein endophilin, which is associated with both CME and CIE undergoes LLPS via both N-BAR-domain driven self-association as well as through SH3 domain-mediated multivalent interactions with its binding partners (Fig. 6), both in the bulk, as well as on membranes. Our results suggest that LLPS could play crucial roles in the formation of endophilin-rich clusters on the plasma membrane, where the clusters serve as initiation sites for FEME. The liquid-like clusters allow for the sorting of activated receptors that also act as multivalent binding partners of endophilin, in a process that leads to the formation of transport carriers[13].

FEME is driven by rapid (within 10 s) assembly and disassembly of more than 10 types of proteins within submicron-sized membrane domains[13,15]. In the biological system, protein assemblies are often regulated by LLPS that forms a condensed phase, known as scaffolds, via multivalent protein–protein or protein–nucleic acid interactions[45]. These phase-separated scaffolds can further concentrate client proteins on the basis of the client's ability to partition into the scaffolds. As we show here, inherent scaffold forming abilities can facilitate protein assembly in endocytic processes by allowing partitioning of its binding partners as clients. Similar to other LLPS systems, we found scaffold–client interactions either to promote demixing or to regulate condensate size depending on the nature of the specific interactions between the client and endophilin[46]. Notably, the promotion of LLPS by two crucial multivalent clients—the C terminal domain of LPD and the TIL of β1-AR suggests that these proteins might be engaged in a switch-like action in endocytosis to initiate protein condensation by shifting the phase boundary to lower endophilin concentrations. It is likely that the liquid-like clusters further facilitate the recruitment of

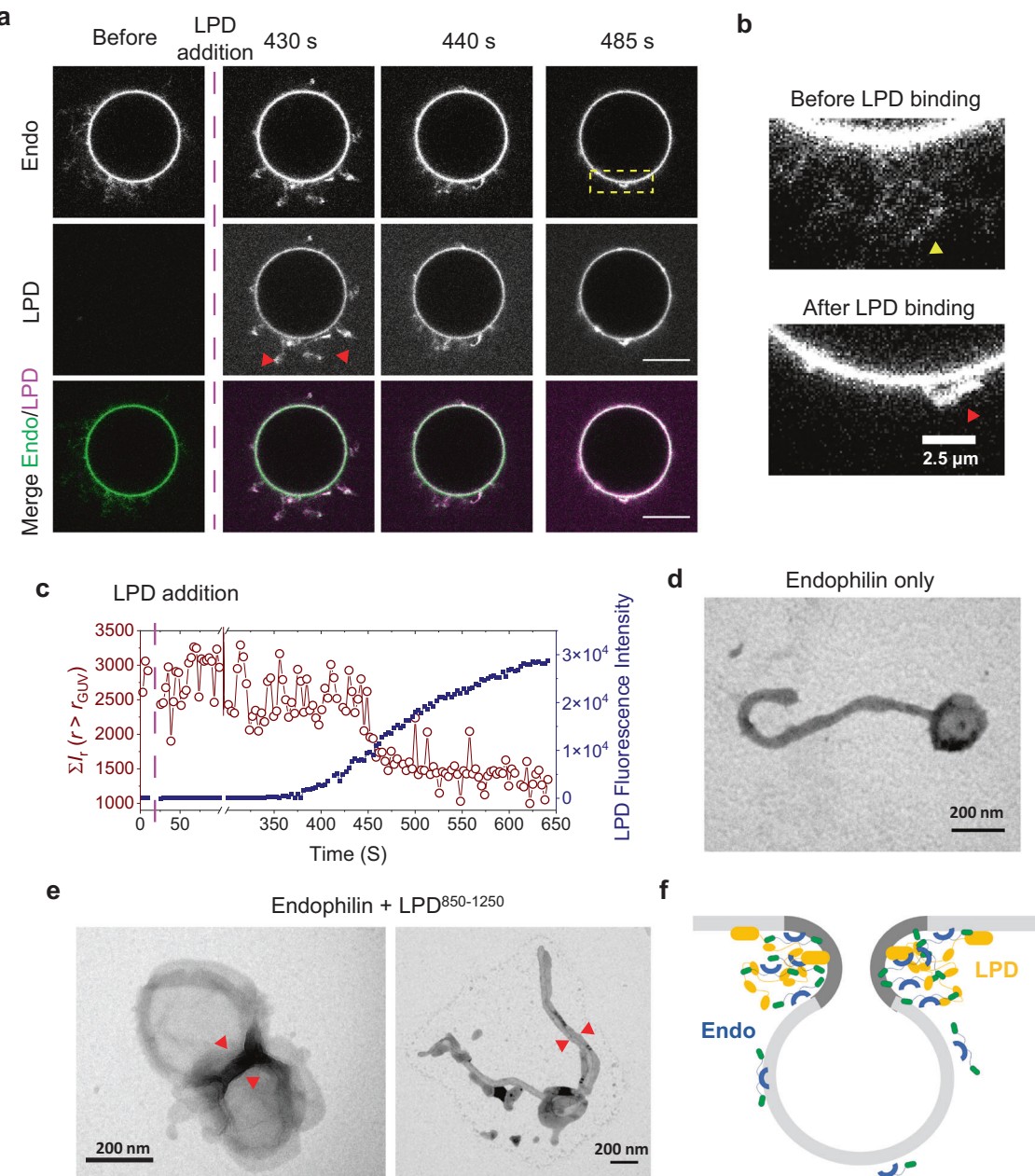

**Fig. 6 | LPD drives membrane adhesion in the presence of endophilin and might stabilize membrane bud formation at FEME sites. a** Time-lapse images of GUVs tubulated in the presence of endophilin before and after the addition of LPD[850–1250] showing recruitment of LPD causes contraction of generated tubules. Top panel, GUVs enriched with endophilin (1 µM, 10% Alexa 488 labeled); middle panel, binding of LPD[850–1250] on the GUVs; bottom panel, endophilin and LPD[850–1250] channels merged. Scale bar 10 µm. **b** Enlarged lipid channel image from the box enclosed region of **a** to show long tubules present in the absence of LPD (yellow arrow) and small structures (red arrow) formed after contraction of tubules upon LPD addition. **c** Quantitative analysis showing changes in tubule length correlates with the extent of LPD recruitment onto GUV membrane. To approximately estimate the tubule length, a radial average of the intensities of the area outside the GUV radius ($r > r_{GUV}$) was estimated in the lipid channel. A sum of average intensities along the radius $\sum_{r > r_{GUV}} \bar{I}_r$ from a single frame was plotted against time (red circles). To estimate LPD binding over time, the fluorescence intensity of Alexa 647 was estimated from the GUV surface and plotted against time (blue squares). **d** Representative TEM image showing membrane tubules generate in the presence of endophilin. **e** Representative images showing that LPD in the presence of endophilin cause membrane adherence. Membrane adherence leads to LUV–LUV adhesion without tubulation (left) and also to tubules adhered along their length (right). All experiments were performed in 20 mM HEPES buffer, 150 mM NaCl, 1 mM TCEP, pH 7.4, and at room temperature. **f** A proposed model showing how endophilin and LPD multivalent interactions might stabilize negative membrane curvature (dark gray area of the membrane) at the neck region of a membrane bud during FEME.

downstream effector proteins such as dynamin, which is known to be recruited by endophilin during FEME to promote membrane scission[15].

The coupling between membrane curvature generation and receptor activation remains a key question in FEME[13,15]. When present on the membrane at a sufficiently high density, endophilin alone can generate membrane tubules via its N-BAR domain[21,47]. However, in

cellulo experiments have shown that membrane invaginations are highly connected to the activation of endocytic pathways[14,69]. In a recent report, Bergeron-Sandoval et al. have proposed that mechanical forces applied on the membrane by viscoelastic protein condensates formed in CME can facilitate membrane deformation[70]. Using endophilin and LPD C-terminal domain reconstituted membrane mimics, we observe here that multivalent interactions can facilitate adhesion of

coated membrane surfaces and the stabilization of negative Gaussian curvature such as is found at membrane budding sites[69].

A current hypothesis suggests that receptor activation enables TIL-mediated recruitment of endophilin to the membrane that possibly enhances the local concentrations of endophilin beyond the threshold required to cause membrane deformation[13,14]. We establish here that TIL itself drives LLPS by multivalent interactions with the SH3 domain of endophilin and promotes clustering of endophilin on the membrane. Within the limitations of our current in vitro model, we propose the following mechanisms of carrier formation during FEME. On one hand, membrane remodeling can occur through enhancement of local endophilin N-BAR activity within the clusters and promotion of membrane scaffolding by rigid N-BAR assembly. On the other hand, TIL-mediated enhancement in endophilin-rich clusters would also allow partitioning of various other proteins, including N-BAR protein BIN1, which recruits downstream proteins such as Dynein to the FEME carriers[50]. Curvature generation could therefore be facilitated by a local enrichment of BAR-proteins at the FEME sites. To understand how and when various other proteins, such as actin regulatory machineries, are engaged to facilitate carrier formation process would require a more rigorous model, ideally using reconstituted full-length proteins. In addition, how the local composition of membrane phospholipids affects these interactions is a key question yet to be addressed. Nevertheless, our study shows many of these interactions could now be understood in the light of protein-protein phase separation driven by multivalent interactions.

To summarize, we have shown that the phase separation of endophilin, combined with the driver-plus-regulator actions of multivalent binding partners of the endophilin-SH3 domain, results in the formation and regulation of endocytic protein complexes. Liquid-like assembly of endophilin and LPD could help accomplish two critical aspects of FEME−(1) dynamic formation of endophilin-rich molecular scaffolds to serve as initial cargo capture sites, and (2) sorting of activated receptors and sequestration of various endocytic proteins via client-partitioning into the scaffolds. Clients, depending on their specific molecular interactions with endophilin on the membrane, can regulate cluster formation efficiencies and cluster size. Additional clustering abilities shown by the TIL could be a driving force for the maturation of transient clusters formed by endophilin–LPD complexes at the FEME initiation sites that would further lead to the formation of stable membrane transport carriers after receptor activation[13–15]. Our findings further suggest that endophilin–TIL interactions enabled by receptor activation is a crucial step in the formation of endocytic protein complexes during FEME. The formation of liquid-like assemblies by multivalent protein-protein interactions might also drive steps downstream of cargo sorting such as engagement of actin regulatory machinery involving N-WASP and Arp2/3[15] and membrane scission via endophilin-dynamin complexes[71]. Here we provide evidence suggesting that the phase separation behavior of endocytic proteins, driven by endophilin self-association as well as multivalent interactions involving both adapters and receptors, could be the mechanistic handle in understanding the formation and regulation of protein assembly in clathrin-independent membrane trafficking.

## Methods

### Chemicals
Alexa Fluor™−488 C5-maleimide, Alexa Fluor™−488 5-SDP ester, Alexa Fluor™−594 C5-maleimide, Alexa Fluor™−633 C5-maleimide, Alexa Fluor™−647 conjugated bovine serum albumin, and Texas Red™ 1,2-dihexadecanoyl-sn-glycero-3-phosphoethanolamine (Texas Red™ DHPE) were procured from ThermoFisher Scientific (USA). Alexa Fluor™−488-α-Synuclein was generously provided by Elizabeth Rhoades's lab. Polyethylene glycol (average MW 8,000) was obtained from Sigma-Aldrich (USA). Tris(2-carboxyethyl)phosphine hydrochloride

(TCEP·HCl) was obtained from AlfaAesar (USA). Sucrose and common reagents for buffer preparation including HEPES, Tris, NaCl, $Na_2HPO_4$ and $NaH_2PO_4$, dithiothreitol (DTT), and ethylenediaminetetraacetic acid (EDTA) were obtained from Fisher Scientific (USA). Lipids 1,2-dioleoyl-sn-glycero-3-phosphocholine (DOPC), 1,2-dioleoyl-sn-glycero-3-phospho-L-serine (DOPS), 1,2-dioleoyl-sn-glycero-3-phosphoethanolamine (DOPE), and 1,2-dioleoyl-sn-glycero-3-[(N-(5-amino-1-carboxypentyl) iminodiacetic acid)succinyl] (nickel salt) (Ni-NTA DOGS) were purchased from Avanti Polar lipids (AL, USA).

### Plasmids
Full-length *rat* endophilin A1 [mutated to a single-cysteine form for labeling, C108S, E241C, C294,295 S], endophilin-N-BAR, and endophilin-dH0 were encoded by plasmids described previously[13]. Plasmids encoding human amphiphysin 1, as well as a truncate in the form of the N-BAR domain were provided by Pietro De Camilli's lab. FBP17 was obtained from Harvey McMahon's lab. The pMal-Abl-PRM5R plasmid was a generous gift from Michael Rosen's Lab (Addgene plasmid #112088). The sequence encoding 7× repeats of a PRM from lamellipodin (aa 970–981) containing (Gly-Gly-Ser)$_4$ linkers and an N-terminal tryptophan was custom synthesized by Biomatik Corporation (Canada). The PRM7 sequence was cloned into the pMal-Abl-PRM5R plasmid by replacing its PRM5 sequence. The C-terminal TEV protease cleavable site between the PRM7 and the 6x His tag was further mutated to stop the cleavage of the His tag during purification. The LPD$^{850–1250}$ (with C-terminal His$_6$ tag), BIN1-isoform 9, and human VASP (wild type) sequences were also synthesized and cloned into a pGEX6p1 vector by Biomatik. TIL of human β1 adrenergic receptor was cloned into a pGEX6p1 vector as described elsewhere;[11] a TIL sequence with a polyhistidine tag was synthesized and inserted into pGEX6p1 vector by Biomatik Corporation (Canada).

### Protein expression and purification
BL21-CodonPlus(DE3)-RIL cells (Agilent Technologies) were transformed with the plasmid of interest. Large volume cultures (2×1 L for TIL and TIL-His, 4×1 L for PRM7-His, Amph1 N-BAR, FBP17, endophilin, endophilin dH0 and N-BAR; 8 L for Amph1) were grown from a starter culture (100 mL), shaking at 225 RPM at 37 °C until O.D. at 600 nm reached 0.5–0.8. Cultures were induced with IPTG (300 μM for TIL and TIL-His, 600 μM for Amph1, Amph1-N-BAR, endophilin, endophilin ΔH0, and N-BAR, 1 mM for FBP17 and PRM7-His) and expression was carried out at 18 °C for -16 h. Cells were harvested by centrifugation at 6000 x g for 10 min, resuspended in a lysis buffer (300 mM NaCl, 50 mM Tris, 2 mM DTT, 1 mM EDTA, pH 8.0) containing 1 mM PMSF. The lysis buffer contained additional 20 mM imidazole but no EDTA for PRM7-His. For Amph1 N-BAR, 10% glycerol was added to the lysis buffer. Bacterial cells were lysed by tip sonication and centrifuged at $30,000 \times g$ for 1 h to remove debris. The supernatant was filtered through 0.22 μm-pore syringe-tip filters (Millipore-Sigma), then purified by FPLC. Protein-specific details of the purification methods are given below.

### TIL and TIL-His
Both TIL and TIL-His were expressed as GST-fusion proteins[71]. Cell lysate was prepared as described above. In short, the GST-tagged protein was purified from the cell lysate by GST Trap affinity chromatography (GE Healthcare). The GST tag was cleaved from TIL by PreScission protease, and the resulting protein mixture was purified by gradient elution from a HiTrap SP HP cation exchange column (GE Healthcare).

### PRM7-His
The PRM7-His construct was expressed as an N-terminal MBP and C-terminal His-tagged fusion proteins. Cell lysate was prepared as described above and loaded to a HisTrap HP affinity column (GE Healthcare) with the help of the EDTA-free lysis buffer containing imidazole as described above. Bound protein was washed with the lysis

buffer and eluted with a high-imidazole elution buffer (300 mM NaCl, 50 mM Tris, 2 mM DTT, 500 mM imidazole, pH 8.0). The elution was loaded to a HiTrap SP HP cation exchange column (GE Healthcare) and eluted over a gradient of NaCl (150 mM NaCl, 20 mM sodium phosphates (monobasic and dibasic), pH 7.0, 1 mM EDTA, 1 mM TCEP; buffer B: 1 M NaCl, 20 mM sodium phosphates (monobasic and dibasic), pH 7.0, 1 mM EDTA, 1 mM TCEP). Fractions containing PRM construct were identified via SDS-PAGE.

PRM7-His fractions were dialyzed against 2×1 L of anion exchange buffer A (150 mM NaCl, 50 mM Tris, 1 mM EDTA, 1 mM DTT, pH 8.0) using a dialysis membrane having 3500 kDa molecular weight cut-off (Fisherbrand) for 12 h, at 4 °C. The dialyzed protein was loaded onto an anion exchange Q HP column (GE Healthcare) and MBP-PRM7-His was collected as flow through. The flow through was cleaved with TEV protease [~1 mg in 15 mL] at 4 °C for ~12 h. Following cleavage, the protein mixture was passed over amylose resin (New England BioLabs). Pure PRM7-His was collected as flow through, concentrated in Amicon™Ultra centrifugal filters (Millipore-Sigma), and exchanged with a HEPES buffer (20 mM HEPES, 150 mM NaCl, 1 mM TCEP, pH 7.4). Concentrations were determined from the tryptophan absorption at 280 nm ($\varepsilon_{280}$ 5500 M$^{-1}$ cm$^{-1}$) using a Nanodrop instrument (Thermo Scientific).

### LPD$^{850-1250}$

Filtered supernatant was incubated with Ni$^{2+}$-NTA Agarose beads (Gold Biotechnology, Inc) overnight at 4 °C. The protein-bound resin was washed with excess lysis buffer and incubated overnight with PreScission protease to remove the GST tag. The bound protein was further washed with lysis buffer and eluted using a high imidazole buffer (150 mM NaCl, 50 mM Tris, 500 mM Imidazole, 2 mM DTT, pH 8). The fractions containing LPD$^{850-1250}$ were identified via SDS PAGE, then combined, and buffer exchanged to 150 mM NaCl, 20 mM sodium phosphates, 1 mM DTT, pH 7.0. The protein was further purified using a HiTrap SP HP cation exchange column using a 150 mM–1 M NaCl gradient (20 mM Sodium phosphate, 1 mM DTT, pH 7). The pure fractions were concentrated and buffer exchanged to 20 mM HEPES, 150 mM NaCl, 1 mM TCEP, pH 7.4.

### N-BAR domain proteins

Full-length endophilin A1, endophilin ΔH0, N-BAR-only, N-BAR-ΔH0, ΔN-BAR mutants of endophilin, and BIN1-isoform 9 were expressed as GST fusion proteins and purified following already established protocols[11,40]. The cell lysate obtained as described above was put through GST-affinity chromatography. After elution was collected, GST tags were cleaved using PreScission protease. Cleaved proteins were further purified by anion exchange chromatography followed by size exclusion chromatography. SDS-PAGE was conducted between chromatography steps to determine protein purity. Final protein concentrations were determined by absorption at 280 nm (via Nanodrop). Purification of Amph1 and its N-BAR domain was performed following the same protocol as described for endophilin with the following modifications: both proteins were purified by cation exchange chromatography after the protease cleavage. Additionally, for Amph1-N-BAR, all buffers used for the purification contained 10% glycerol.

### FBP17

Cell lysate was prepared as described above and loaded to a GSTrap™FF column (GE Healthcare) in lysis buffer (300 mM NaCl, 50 mM Tris, 2 mM DTT, 1 mM EDTA, pH 8.0). The column was washed with a wash buffer (150 mM NaCl, 50 mM Tris, 2 mM DTT, 1 mM EDTA, pH 8.0) and the protein was eluted with a glutathione-containing buffer (150 mM NaCl, 50 mM Tris, 2 mM DTT, 1 mM EDTA, 20 mM glutathione, pH 8.0). Eluted protein was simultaneously digested with PreScission protease and dialyzed across a membrane (Fisherbrand) into a glutathione-free buffer (150 mM NaCl, 50 mM Tris, 1 mM EDTA, 1 mM DTT, pH 8.0). Dialyzed and cleaved protein was finally purified by size exclusion chromatography in a HEPES buffer (150 mM NaCl, 20 mM HEPES, 1 mM TCEP, pH 7.4).

### Fluorescent Labeling of proteins

TIL and TIL-His were labeled at the N-terminal with Alexa Fluor™−488 5-SDP ester; LPD$^{850-1250}$ was labeled with Alexa 647 succinimidyl ester; protein was exchanged to an amine-free buffer (20 mM sodium phosphates, 150 mM NaCl, pH 7.0) and the label was added to the protein in 5 times molar excess and allowed to react for 24–36 h at 4 °C. FBP17, Amph1, PRM7-His, endophilin, and endophilin truncations were labeled with a 5C-maleimide-linked fluorophores, with label added to the protein at a 5 times molar excess and allowed to react for either 4 h (FBP17) or 12–16 h (PRM7-His, Amph1, endophilin, and its mutants). In all cases, the labeled protein was separated from an excess dye by passing over HiTrap™ desalting columns (2×5 mL) (Cytiva).

### Protein concentration determination after labeling

For endophilin and its mutants, Amph1, FBP17, PRM7-His, and LPD$^{850-1250}$, the protein absorbance at 280 nm and the fluorescent label absorbance at the corresponding wavelengths were measured using a Nanodrop instrument. The contributions from the fluorophore to the absorption at 280 nm were determined from the absorption spectra of protein-free fluorophores and were subtracted from the absorbance of the labeled protein to determine the protein concentration accurately.

TIL and TIL-His concentrations could not be determined from absorption at 280 nm, since they have no tryptophan. First, the concentration of fluorescently labeled protein was determined from the fluorophore absorbance using a Nanodrop instrument. In order to obtain the total protein concentration, the mixture of labeled and unlabeled proteins was injected onto a ZORBAX Eclipse XDB C8 HPLC column (Agilent, USA) and eluted with a gradient of acetonitrile and water (Solvent A: water and 0.1% trifluoroacetic acid, Solvent B: acetonitrile with 0.1% trifluoroacetic acid) using an LC-10AT solvent delivery systems (Shimadzu Corporation). Chromatograms at 220 nm (peptide) and 340 nm (for the secondary absorption band of Alexa 488) were recorded via SPD-10A detector (Shimadzu Corporation). The labeled and the unlabeled proteins appeared as two different peaks in the chromatogram. The relative concentrations of the labeled and unlabeled proteins were determined from the ratio of the area under the two peaks. The concentration of the unlabeled protein was estimated from the ratio and the concentrations of the labeled protein obtained via Nanodrop measurements.

### Construction of phase diagrams

Phase diagrams were built by scoring protein mixtures at the given protein and PEG concentrations for the presence of droplets with transmitted light microscopy. For these experiments, protein solutions (10 μL) in HEPES buffer were visualized in a 384-well plate (Greiner Bio-One, Austria) with an Olympus IX71 inverted microscope using a ×40 0.75 NA air objective (Olympus, Center Valley, PA).

### Confocal microscopy

Confocal imaging was performed using an Olympus IX83 inverted microscope equipped with FluoView 3000 scanning system (Olympus, Center Valley, PA). Images were taken at room temperature using a ×60 1.2 NA water immersion objective (Olympus). Imaging of multi-protein systems was performed via orthogonal fluorescent labeling of proteins using Alexa 488 ($\lambda_{ex}$ 488 nm, $\lambda_{em}$ 500–540 nm), Alexa 594 ($\lambda_{ex}$ 561 nm, $\lambda_{em}$ 580–620 nm), and Alexa 633/Alexa 647 ($\lambda_{ex}$ 640 nm, $\lambda_{em}$ 650–700 nm). The excitation lasers were alternatively used to minimize cross-talk between the different channels using a sequential line scan mode. Images were analyzed with ImageJ (version 1.52a), MATLAB (versions R2019b and R2020a), and Python (version 3.7.1) programs.

For imaging droplets in solution, glass coverslips (25 × 25 mm², Fisher Scientific) passivated with BSA (2 mg/mL in HEPES buffer) were

used. Solutions (5–10 μL) containing the droplets were imaged in a closed chamber created by sandwiching two coverslips ($25 \times 25$ mm$^2$, Fisher Scientific) using vacuum grease. For supported bilayers, commercially available glass-bottomed chambers (see Planar Supported Bilayer Preparation section) were used for imaging.

## Fluorescence recovery after photobleaching (FRAP) studies on droplets

For FRAP studies, a circular region of interest (ROI) within a protein condensate settled on the glass coverslip was bleached using short exposures (~ 500 ms) of 488 nm or 561 nm or 640 nm lasers at 100% laser power. The ROI size and the bleach time were adjusted to keep the bleached area ≤20% of the droplet area. The collection of images for the recovery stage was started immediately after the bleaching.

FRAP data were analyzed using ImageJ (for image intensity extraction) and Microsoft Excel (for quantitative analysis). For each time frame, mean intensities were estimated for an ROI within the bleached region and of another ROI from the unbleached region. The ratio of intensities at bleached ($I_{bleach}(t)$) and unbleached regions ($I_{unbleach}(t)$) were determined at each time point which was further normalized to 1 for the intensity ratio before photobleaching ($q(t_{prebleach})$) and to 0 for the intensity ratio at 0 s after photobleaching ($q(t_0)$) using the following formula:

$$I_{norm}(t) = \frac{q(t) - q(t_0)}{q(t_{prebleach}) - q(t_0)} \qquad (1)$$

$$\text{where } q(t) = \frac{I_{bleach}(t)}{I_{unbleach}(t)}$$

The normalized intensities were plotted against time to obtain a fluorescence recovery profile. The recovery profile was fit to either a single exponential model,

$$I(t) = A(1 - e^{-\frac{t}{\tau}}) \qquad (2)$$

or a double exponential model, $I(t) = A\left[ B\left(1 - e^{-\frac{t}{\tau_1}}\right) + (1 - B)\left(1 - e^{-\frac{t}{\tau_2}}\right) \right]$

$$(3)$$

as described elsewhere[37]. $A$ represents the mobile fraction and $B$ represents the fractional contribution of the time component $\tau_1$. The values of $A$ and $B$ were limited to the interval [0,1] and time constants ($\tau$, $\tau_1$, or $\tau_2$) were limited to the interval [$t_{frame}$, ∞] where $t_{frame}$ is the spacing between frames in a given FRAP experiment. The average photorecovery halftime ($t_{1/2}$) was calculated by multiplying either $\tau$ or $\tau_{av}$ (where $\tau_{av} = B\,\tau_1 + (1 - B)\,\tau_2$) with the natural logarithm of 2.

The image acquisition frequencies were adjusted according to the measured recovery rates (e.g., 3 s per frame for TIL/endophilin droplets, whereas 0.17 s per frame for PRM7/endophilin droplets) so that sufficient data points are collected within the rise time of the recovery profile. During exponential fitting, the lower bound of the time constant was set to the time resolution of our image acquisition so that the fitting would not result in a value lower than what we could measure.

## Determination of apparent partition coefficients

Partition coefficients were determined following an earlier reported method[45]. A typical experiment consisted of preparing a 10 μL suspension of endophilin (25 μM) droplets in the presence of 10% PEG. The partitioning proteins (clients) labeled with an Alexa™ fluorophore were mixed with endophilin, at a concentration between 250 nM and 1 μM, prior to inducing the droplet formation by adding PEG. Droplet samples for confocal imaging were prepared as described above. The photodetector settings were optimized to ensure both bulk and

droplet fluorescence intensities were above the background but below the saturation level of the detector. Images were collected at the equatorial plane of the droplet and 10 and 20 droplets were imaged per experiment. Fluorescence intensities at the interior of droplets and in the bulk solution surrounding them were obtained using ImageJ. The background intensity was estimated by imaging a solution containing no fluorescent protein under identical imaging conditions. The apparent partition coefficient is defined as the ratio of the background subtracted intensities inside and outside the droplet.

## Determination of volume occupancies of PEG and disordered proteins

Volume occupancy ($\varphi$) was defined as, $\varphi = cv$ where, $c$ is the concentration of the component and $v$ is its specific volume[30]. For PEG, the literature reported value of $v$, 0.84 mL/g was used[30]. For TIL, LPD$^{850-1250}$, and PRM7, $v$ was estimated from their hydrodynamic radii ($R$) using the formula[72],

$$v = \frac{4\pi R^3 N_A}{3M}$$

where, $N_A$ is the Avogadro's number and $M$ is the molecular weight. We estimated the $R$ values for TIL, LPD$^{850-1250}$, and PRM7 as 30, 62, and 40 Å, respectively, from their sequences using a formula established by Marsh and Forman-Kay[73].

## Planar supported bilayer preparation

Liposomes were prepared following established protocols[74], with minor modifications. A lipid mixture containing DOPC and Ni$^{2+}$-NTA-DOGS (99:1) was dried under a stream of nitrogen, and vacuum dried for a time period between 2 and 16 h. The lipid film was rehydrated in HEPES buffer with a final concentration of 1 mg/mL, by vortexing for 1 min in 10 s intervals. The solution was sonicated for 30–40 min at room temperature and subsequently subjected to five rapid freeze–thaw cycles. Finally, the LUV suspension was extruded 15 times through a polycarbonate membrane of 100 nm pore size.

Supported bilayers were prepared following a method described elsewhere[24], with minor modifications. Briefly, chambered glass coverslips (Lab-tek, Cat #155409) were cleaned by rinsing with 50 % (v/v) isopropanol, followed by incubation in 10 M NaOH for 2 h, and finally washed thoroughly with Milli-Q water. A freshly prepared LUV suspension was added to the coverslips and incubated for 30 min (150 μL, 0.5–0.7 mg/mL). The wells were washed with HEPES buffer (3 ×450 μL) to remove intact, unadsorbed liposomes. The wells were incubated with equal volume BSA (fatty acid free) solution (2 mg/mL, in HEPES buffer) for 30 min, and washed with HEPES buffer (3 ×450 μL). The bilayers were immediately used for protein conjugation.

## Protein conjugation to the planar-supported bilayer

For single protein (with His$_6$-tag) coupled bilayers, PRM7 (500 nM) or TIL (500 nM) or LPD$^{850-1250}$ (50 nM) was added to the solution and gently mixed. The protein was incubated for 15 min and was gently rinsed with HEPES buffer (2×600 μL). After protein conjugation, the bilayers were allowed to rest for 30 min for equilibration before subsequent experiments.

Bilayers coupled with two proteins were formed by sequential coupling of PRM7 followed by TIL. PRM7 (250 nM) was first incubated with the bilayer as described above and allowed to rest for 15 min. TIL (50 nM) was then introduced to the solution and gently mixed. After 5–10 min of incubation, the solution was gently rinsed with HEPES Buffer (2×600 μL) and the bilayer was allowed to rest for 15 min. This method ensured substantial and uniform fluorescence intensity of both proteins on the bilayer. Higher protein concentrations and longer incubation periods led to the formation of protein clusters on the bilayers. The alternative approach of simultaneously adding equimolar

(250 nM) PRM7 and TIL resulted in starkly different fluorescence intensities of the two proteins on the membrane, potentially due to differences in their association kinetics.

FRAP was performed on the bilayers to confirm the fluidity of the protein(s) and the bilayer. All experiments were performed within 4 h of depositing LUVs on the coverslips.

## Estimation of TIL densities on the planar-supported bilayers

TIL densities on the supported bilayers were determined from the fluorescence intensities of Alexa 488-labeled TIL following an earlier established method[11,24]. To calibrate the fluorescence intensity via the mol% of membrane-bound fluorophore, supported bilayers composed of varying concentrations of BODIPY-FL-DHPE were prepared. Two different LUV preparations, containing 0% and 0.75% BODIPY-lipid, respectively, along with DOPC, were mixed in three different ratios and then incubated on chambered glass coverslips to form supported bilayers containing 0.25%, 0.5%, and 0.75% BODIPY-lipid. Calibration curves were generated using identical instrument settings used for imaging TIL-Alexa 488 on the bilayers. Assuming the surface area of a lipid headgroup 0.7 nm$^2$ and considering Alexa-488 a two times brighter fluorophore than BODIPY as reported previously[24], TIL-Alexa 488 densities were determined to be $330 \pm 15$ in the case of bilayers containing of TIL-only and $140 \pm 20$ in the case of bilayer where both TIL and PRM7 were lipid coupled.

## Auto-correlation and cross-correlation analysis

Homebuilt analysis methods written in MATLAB (version R2020a) and Python (version 3.7.1) were used to perform the correlation analysis and radial averaging. The *normxcorr2* function implemented in MATLAB was used to perform the auto and cross-correlation of images. The function takes two images as inputs, a template image, and the input image. Depending on the choice of the inputs, the function can be used to determine the auto and cross-correlation of images. The output of the analysis was a normalized 2-D correlation matrix with values ranging from [−1,1]. The correlation matrix obtained was radially averaged in increments of 1 pixel with respect to the matrix's center, thus providing us with radially averaged normalized auto and cross-correlation data[24]. The normalized autocorrelation plots were fitted with a single-exponential function $G(r) = A \ e^{-r/R}$ to obtain correlation length ($R$). As cross-correlation is not in general commutative, the correlation analysis for 1-2 and 2-1 was performed where 1 and 2 refer to the two images being analyzed. However, we found that the cross-correlation plots for 1-2 and 2-1 were completely overlapping (Supplementary Fig. 13d) and therefore reported one of the curves only.

## GUV preparation and imaging

GUVs were prepared by the electroformation method using indium tin oxide (ITO)-coated slides[11]. Chloroform solutions of desired lipid composition were coated onto ITO slides and vacuum dried for at least 2 h. The lipid films were hydrated with sucrose solution (350 mOsm in MilliQ purified water). Electroformation was performed at 55 °C for 1 h.

For phase separation experiments, GUVs composed of DOPC and Ni$^{2+}$-NTA-DOGS (99:1) containing 0.2 mol% Texas Red were mixed with LPD$^{850-1250}$-Alexa 647 (100 nM) in HEPES buffer (in 1:40 GUV solution: buffer volumetric ratio) and incubated at room temperature for 10 min. The buffer osmolality was pre-adjusted to the osmolarity of the sucrose solution used for electroformation. Followed by the incubation, endophilin-Alexa 488 (200 nM) was added to the mixture. For imaging, the mixture was transferred into a Lab-tek chamber whose surface was already passivated by BSA (2 mg/mL).

For tubulation studies, GUVs composed of DOPS/DOPE/DOPC/ Texas Red (45:30:24.8:0.2 molar ratio) were mixed (in 1:40 GUV solution: buffer volumetric ratio) with endophilin (1 μM, containing 10% Alexa 488 labeled endophilin) in HEPES buffer (osmolality adjusted). The mixture was transferred to a Lab-tek chamber whose surface was

covered by pre-formed solid-supported bilayers composed of DOPC, to prevent adhesion of proteins and GUVs to the bottom surface. GUVs that contained tubules formed by endophilin were located at the bottom of the chambers and imaged at their equatorial plane. LPD$^{850-1250}$-Alexa 647 was added to the chamber solution (to a final concentration of 100 nM) and allowed to diffuse to the bottom of the chamber, causing minimal physical perturbation to the GUV under focus. Time series were recorded at the rate of 3.2 s per frame imaging speed.

## Fluorescence intensity analysis on GUVs

Fluorescence intensity on the perimeter of GUVs was estimated using a homebuilt analysis method implemented in Python (version 3.7.1) and the computer vision package OpenCV (Version 4.4.0.44).

## TEM imaging on LUVs

TEM imaging of LUVs was performed following a method described elsewhere[11]. LUVs of lipid composition DOPS/DOPE/DOPC (45:30:25) were prepared following the method described under "solid supported bilayer preparation," with the following modifications: no freeze–thaw cycles were performed and 400 nm pore size membranes were used for extrusion. For tubulation studies, LUVs (0.1 mM total lipid conc.) were incubated with endophilin (10 μM) in HEPES buffer for 20 min at room temperature before loading onto TEM grids. LPD$^{850-1250}$ was added (to a final concentration of 400 nM) to the endophilin-LUV mixture right before the grid preparation. Samples were applied to the grids immediately after LPD$^{850-1250}$ addition since incubation for longer than 5 min caused the formation of large, micron-sized clusters. Samples were allowed to adhere on the grids for 2 min and excess liquids were soaked by Whatman papers. Grids were washed 3 times with HEPES buffer, after each wash the excess liquid was removed by Whatman papers. For staining, 2% uranyl acetate solution was used for 2 min. Extra stains were soaked by Whatman papers and the grids were washed thrice with MilliQ water followed by drying on a filter paper for 10 min at room temperature. Images were recorded on a JEM 1011 transmission EM (JEOL, USA), operated at 100 kV, coupled with an ORIUS 832.10 W CCD camera (Gatan). Post-processing of images was performed with ImageJ software (Version 1.52a).

## Statistics and reproducibility

For all graphs where statistical analyses were applied, the number of repeats ($N$) has been mentioned in the figure legends. Data shown as representative images are also repeated multiple times to ensure reproducibility. Droplet formation experiments (Figs. 1b and 3b and Supplementary Figs. 1a, b, 4, 5a–c, and 11) were repeated at least three times with similar results every time. All FRAP experiments on droplets (Figs. 1c, 2d, and 3c–e and Supplementary Figs. 7a and 10) were repeated three times. Client partitioning into droplets (Fig. 2b and Supplementary Figs. 6a and 8a), Amph1 and BIN1 partitioning (Fig. 2f and Supplementary Fig. 8b), and droplet size regulation (Fig. 2h) experiments were reproduced thrice. Protein clustering on supported bilayer experiments (Figs. 4c, d and 5a, d, g and Supplementary Fig. 12a-c) were repeated at least in triplicate with similar results. Experiments on GUVs (Fig. 6a, b and Supplementary Fig. 14) and LUVs (Fig. 6d, e and Supplementary Fig. 15b) were also repeated at least three times with similar observations.

## Reporting summary

Further information on research design is available in the Nature Research Reporting Summary linked to this article.

## Data availability

All processed data generated and analyzed for this study are included in the article and Supporting Information. Raw image files have been uploaded to the Figshare repository [https://doi.org/10.6084/m9.figshare.20388570.v1]. Source data are provided with this paper.

## Code availability

The MATLAB and Python-based codes for correlation analysis and GUV analysis are uploaded to the Github repository [https://github.com/Baumgartlab/Correlation-analysis-and-GUV-analysis].

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

## Acknowledgements

This research was financially supported by the National Institutes of Health, grant No GM 097552. The authors acknowledge Dr. Yi-Wei Chang for valuable discussions and Dr. Michael Rosen for valuable technical information about the purification of multimeric PRMs. The authors are also grateful to Tatyana Svitkina lab for use of the TEM facility and to Dr. Andrea Stout for technical help with STED imaging.

## Author contributions

S.M. and T.B. conceived the project, designed the experiments, and wrote the manuscript. S.M., S.B., K.N., I.P., J.Z., and H.P.J. performed experiments. R.J. helped with protein design and purification. S.M., S.B., and K.N. analyzed the data. K.N. and S.B. contributed to the "Methods" section and edited the manuscript.

## Competing interests

The authors declare no competing interests.
