## [Peer Review File · Nature Communications]

REVIEWER COMMENTS

Reviewer #1 (Remarks to the Author):

The manuscript by Mondal et al demonstrates that endophilin, a major regulator of Fast Endophilin Mediated Endocytosis (FEME), can form macromolecular condensates through liquid-liquid phase separation (LLPS). Purified endophilin can form droplets without any other proteins, at high concentrations and supported by macromolecular crowding via PEG, and this behavior seems to be mediated solely by the endophilin BAR domain. Several endophilin binding proteins (TIL, PRM7, PBP17, and amphiphysin) were shown to partition strongly into endophilin droplets, with some of these affecting condensate-forming propensity. Crowding-independent droplets can be supported by addition of endophilin “clients”. Finally, some of these similar effects were investigated on supported bilayers. The experiments are well conducted and reported, and the results are convincing. However, this reviewer does not agree that the results provide the “mechanistic insights into the priming and initiation steps of FEME” that the manuscript claims.

So many proteins have now been shown to undergo various phase transitions in vitro, that it is no longer surprising or interesting when a protein forms some sort of condensate under certain conditions. Fewer of these observations have been shown to be relevant in living cells. However, these are interesting phenomena with potential significance in various aspects of cellular physiology. The observations and characterizations of endophilin condensates add to this rapidly expanding field, but are relatively limited in scope and potential physiological relevance. There is no attempt to experimentally connect the observations to living cells and the conditions (ie protein concentration and PEG mediated crowding) are difficult to reconcile to what might be relevant for the cytoplasm. Even the in vitro studies are somewhat limited in scope and insight: how do the condensates affect the lipid membrane? How are they related to membrane endocytosis? Overall, while this topic is of potential interest, the present work is somewhat superficial: the observations contribute to the rapidly expanding knowledge base of proteins that show interesting LLPS-like phenomena, but in this reviewer’s opinion, do not contain either sufficient physical insight or physiological significance for Nature Communications.

Other significant issues:

1. Why are the membrane condensates so different from solution? This would seem a highly relevant and interesting question that is largely ignored. Does the membrane inhibit condensate formation? If so, why? Why weren’t these experiments conducted on GUVs to test if condensates bend the membrane or facilitate invagination?
2. There are several speculations about the relevance of these results for regulation of FEME or endocytic protein assemblies, but these cannot go beyond speculations, because the relevance of these structures themselves has not been supported.

3. For the phase diagrams, it is difficult to get a sense of the variance in these experiments. Are different experiments (and especially different preparations) as tight as the phase diagrams would appear to suggest? Put another way, what is a meaningful difference in these experiments? And how would one know whether such differences are meaningful in a biological (rather than statistical) sense?
4. The notion that the intrinsically disordered linker of endophilin inhibits phase separation is interesting and surprising, in light of the many examples of IDPs forming condensates. This could have been another aspect for new insights. And another: what differences between amphiphysin and FBP17 (both BAR-containing proteins) lead to one forming solid-like shell and suppresses LLPS while the other does not?
5. LLPS is clearly an inappropriate description here because of how non-liquid some of the behaviors appear.
6. LPD appears to be a major regulator of endophilin and FEME. It would seem valuable to extend the studies to full-length LPD rather than PRM7.

Reviewer #2 (Remarks to the Author):

Mondal, Baumgart and colleagues are reporting that protein-protein phase separation driven by multivalent interactions plays a key role in the initial steps of Fast Endophilin-mediated endocytosis (FEME) carrier formation.

FEME is a portal of entry into cells that is not always active (unlike for example, Clathrin-mediated endocytosis that is constitutive). FEME regulates key physiological functions (e.g. cell migration, rapid receptor uptake, cell signalling modulation), and its inhibition was recently shown to boost anti-cancer treatments using antibody-dependent cellular toxicity (ADCC)-mediating therapeutic antibodies (Chew HY et al. Cell 2020).

They provide evidence that i) the BAR domain of Endophilin drives liquid-liquid phase separation in crowded environment, ii) Endophilin binding partners (cargo receptor tail (bet1-Adenergetic receptor third intracellular loop (beta1-AR TIL) and accessory proteins Lamellipodin proline-rich motifs (Lpd PRM), FBP17 and Amphiphysin 1) partition into LLPs droplets, and iii) that such client proteins regulate LLPS formation depending on their molecular features. Then, the authors demonstrated that, iv) Endophilin undergoes LLPS upon binding to multiple PRM containing ligands even in the absence of crowding agents and, v) causes clustering of membrane bound beta1-AR TIL and Lpd PRM by two-dimensional phase separation. Interestingly, they report that Lpd PRMs pre-enrich Endophilin on membrane but only the addition of TIL triggers robust cluster formation.

This is a very important manuscript as it constitutes the first in vitro reconstitution of the priming and initiation steps of FEME and it recapitulate the steps observed in cells in a controlled environment. And importantly, it reveals the molecular mechanism by which FEME carriers are initiated and demonstrates how multivalent interactions between the key initial proteins drive they local concentration up to a phase transition into liquid-like condensates. This manuscript is an important addition to our understanding of how FEME works and complement the existing literature.

The findings may well prove to apply to other endocytic and intracellular membrane trafficking steps.

The experiments were carefully designed (with appropriated controls), executed and analysed and constituted a high-quality piece of work.

My only comments are the following:

1) The authors report the very interesting finding that Amphiphysin 1 does not form LLPS droplets (Sup Fig. 5) on its own and is recruited into a solid-like shell around Endophilin driven LLPS droplets (Figure 2F-H).

However, Amphiphysin 1 is tissue-restricted and mostly (if not only) expressed in the brain, whereas FEME was also observed in a variety of tissue-culture as well as primary cells from various non-brain organs (kidney, retina, skin and umbilical cord veins). Instead, Amphiphysin 2 (aka Bin1) is ubiquitously expressed, although the expression pattern of its isoforms vary.

Relevant to this work, Ferreira APA et al Nat Comm 2021 recently reported that the ubiquitously expressed isoform 9 of Amphiphysin 2 (Bin1 iso9) functions in FEME by recruiting Dynein onto FEME carriers. The cellular depletion of all Amphiphysin 1 and 2 isoforms did lead to a dysregulation of Endophilin spots, which was attributed to the failure to recruit Dynein (Ferreira APA et al Nat Comm 2021) but the hypothesis from Mondal and colleagues (the present manuscript) that it would also be due to the loss of the regulation of the size of the priming patch by the absence of Amphiphysin could also contribute to the phenotype.

Importantly, the two main Amphiphysin 2 isoforms (iso 9 and 10) that are ubiquitously expressed have much sorter linker between their BAR and SH3 domains and thus resemble much more to Endophilin than to Amphiphysin 1.

Whereas both Amphiphysin 1 and the long isoforms of Amphiphysin 2 bind to Clathrin and AP-2 (the motifs are in the exons skipped in the short isoforms 9 and 10) and function in CME.

Thus, it would be important to complement Figure 2 and Supplementary Figure 5 with one of these short ubiquitously expressed Amphiphysin 2 isoform (aka Bin1 iso 9 or 10) to validate the authors' conclusion that Amphiphysin forms a solid-like shell and antagonise LLPS of Endophilin because of its long disordered linker.

Such experiment is worth doing as both outcome would be informative:

Should the short Amphiphysin 2 isoforms be recruited into LLPS and Amphiphysin 1 (and thus likely the long Amphiphysin 2 isoforms) are excluded, it would confirm the authors' hypothesis that the longer linkers drive the exclusion and an argument could be made for the size control by the long Amphiphysin isoforms.

Should the short Amphiphysin 2 isoforms be also restricted to the periphery of the LLPS, it would provide a basis for the proposition that the exclusion of Amphiphysin 2 (Bin1) to the rim of the forming FEME priming patches might help the recruitment of Dynein to the right location (the periphery of the patches) to apply the pulling tension and trigger the swift FEME carrier formation.

2. The findings may well prove to apply to other endocytic and intracellular membrane trafficking steps, but the current data does not support the broad title: "[...] involved in Clathrin-independent endocytosis [...]" (lines 1-2), as it solely focused on FEME.

Clathrin-mediated endocytosis is broader than FEME and comprises processes as mechanistically different as the micropinocytosis, CLIC/GEEC, Massive endocytosis (MEND), EGFR non-Clathrin endocytosis (EGFR-NCE) and pathways operating according to the GL-Lect hypothesis, for which the mechanisms here may or may not apply.

I would not think that a title restricted to "[...] involved in Fast Endophilin-Mediated Endocytosis [...]" would decrease in any way the importance of this study. As mentioned above, FEME is involved in important physiological processes and its manipulation may have useful therapeutic applications.

3. The text reports a "significant drop in the number of large droplets" (line 336) but statistics are missing on the corresponding panel (Supplementary Fig. 8B)

Reviewer #3 (Remarks to the Author):

Several recent studies have identified roles of liquid-liquid phase separation (LLPS) in endocytic processes. In this work, Mondal et al examine molecular interactions involved in Fast Endophilin Mediated Endocytosis (FEME), a clathrin-independent endocytic pathway. The authors demonstrate that endophilin, a key player in FEME, can undergo LLPS in vitro in the presence of crowding agents. Multivalent binding partners of endophilin that are involved in FEME – including the third intracellular loop (TIL) of the beta-1-adrenergic receptor (a G-protein coupled receptor, GPCR), as well as the proline-rich motifs (PRM) of lamellipodin (LPD) – can trigger LLPS (in the absence of crowding agents) in bulk solutions and on membranes. On membranes, TIL has a higher propensity to phase separate with endophilin, compared to PRM7.

Previously, the molecular mechanism was unknown as to how TIL and endophilin interact to facilitate FEME. This paper suggests that LLPS could play an important role in FEME, with endophilin clusters acting as priming sites for FEME through the recruitment of GPCRs and other endocytic proteins. These clusters could enhance protein activity at the membrane and lead to formation of FEME transport carriers. This work will contribute to the emerging role of LLPS in endocytosis and will be of interest to the membrane biophysics and LLPS communities. I do, however, have several major and minor suggestions for revision:

Major:

1. The authors' interpretation of the FRAP data (Fig. 1) is that endophilin-rich droplets have a biphasic state, with a mobile phase and immobile phase. Are there other biophysical measurements that could back up this claim?
2. The authors did a good job describing the importance of the BAR domain to LLPS of endophilin. I do wonder about one variant that was not tested: What is the phase behavior of just the BAR domain, with both H0 and SH3 deleted?
3. Can the authors further comment on whether PRM7 is a good model for the C-terminal region of LPD, and why they chose this model? The C-terminal domain of LPD is much longer and contains 10 PRMs.
4. Fig. 2F, G are very interesting and indeed suggest that amphiphysin localizes at the interface of endophilin condensates, like a surfactant. However, is the size data in Fig. 2H due to amphiphysin (Amph) acting as a surfactant that stabilizes smaller droplets, or is Amph inhibiting LLPS? In other words, is the total volume of the condensed phase approximately conserved with vs. without Amph?
5. In Fig. 3C, D: It would have been preferable if the authors used the same fluorophore to label endophilin. Also, the recoveries of PRM and TIL are so fast relative to the image acquisition frequency that that it's unclear the fits are quantitatively accurate. It would be helpful if the authors can comment on these issues.
6. It is important to note what buffer conditions (in addition to PEG) were used for these experiments, since ionic strength, pH, etc. are important regulators of LLPS. Although such information is provided in

the Methods, the buffers used in the experiments should also be clearly stated in the captions. For example, this seems quite important for Fig. 3.

7. How do the authors reconcile what appears to be different trends between Fig. 5 (TIL has stronger clustering than PRM7 on the membrane) vs. Fig. 3 (endophilin-PRM system requires lower concentrations for LLPS than endophilin-TIL)?

8. The authors note in the discussion that their lab is currently working to understand how endophilin-TIL interactions ultimately affect membrane curvature and trafficking. Do the authors have any data suitable for sharing at this time to address these tantalizing questions?

Minor:

A. Line 38: Insert comma after “protein”

B. Line 54: Did the authors intend to write “membraneless organelles,” instead of “membrane organelles”?

C. Line 60: Provide the in-text citation to the Day et al paper (ref. 57).

D. Line 126: It would be helpful to provide a conclusion to this paragraph – I believe the authors are saying that the CD data suggests that PEG and LLPS do not cause protein conformation changes.

E. Supplementary Fig. 3B: It may be helpful to add statistical testing here.

F. Line 144: Comma should be changed to period.

G. Figure 1E is missing an x-axis label.

H. Lines 165-169: What data or figure is this text referring to? Is it Supplemental Fig 8A?

I. Given the importance of the radial average autocorrelation function in the analysis here, it would be useful to elaborate a bit further on this method (even briefly), beyond just citing reference 55.

Reviewer #1 (Remarks to the Author):

The manuscript by Mondal et al demonstrates that endophilin, a major regulator of Fast Endophilin Mediated Endocytosis (FEME), can form macromolecular condensates through liquid-liquid phase separation (LLPS). Purified endophilin can form droplets without any other proteins, at high concentrations and supported by macromolecular crowding via PEG, and this behavior seems to be mediated solely by the endophilin BAR domain. Several endophilin binding proteins (TIL, PRM7, PBP17, and amphiphysin) were shown to partition strongly into endophilin droplets, with some of these affecting condensate-forming propensity. Crowding-independent droplets can be supported by addition of endophilin “clients”. Finally, some of these similar effects were investigated on supported bilayers. The experiments are well conducted and reported, and the results are convincing. However, this reviewer does not agree that the results provide the “mechanistic insights into the priming and initiation steps of FEME” that the manuscript claims.

Comment 1

So many proteins have now been shown to undergo various phase transitions in vitro, that it is no longer surprising or interesting when a protein forms some sort of condensate under certain conditions. Fewer of these observations have been shown to be relevant in living cells. However, these are interesting phenomena with potential significance in various aspects of cellular physiology. The observations and characterizations of endophilin condensates add to this rapidly expanding field, but are relatively limited in scope and potential physiological relevance. There is no attempt to experimentally connect the observations to living cells and the conditions (ie protein concentration and PEG mediated crowding) are difficult to reconcile to what might be relevant for the cytoplasm. Even the in vitro studies are somewhat limited in scope and insight: how do the condensates affect the lipid membrane? How are they related to membrane endocytosis? Overall, while this topic is of potential interest, the present work is somewhat superficial: the observations contribute to the rapidly expanding knowledge base of proteins that show interesting LLPS-like phenomena, but in this reviewer’s opinion, do not contain either sufficient physical insight or physiological significance for Nature Communications.

Our response

We thank the reviewer for their critical feedback and for highlighting the limitations of our present study. Our work, as pointed out by reviewers #2 and #3, is the first *in vitro* model of FEME and shows that protein phase separation driven by multivalent interactions could be a key mechanism that drives protein assembly in this clathrin independent endocytosis pathway. In addition, for the first time, we show that BAR-domain proteins can undergo phase separation by homotypic interactions in the presence of molecular crowders, mediated by the folded BAR-domains. With the revised version of our manuscript we provide additional new physical and biologically relevant insights –

- 1) We now show that the natural lamellipodin (LPD) C-terminal region undergoes LLPS upon multivalent interactions with endophilin both in solution and on the membrane. This data strengthens our earlier insight, which was based on the PRM7 model, that multivalent interactions between endophilin and LPD drive protein-clustering and that this can facilitate FEME priming site formation.

- 2) We show that multivalent protein assembly on the membrane drives membrane adhesion and can regulate endophilin mediated membrane curvature. Based on our observations on model membranes such as giant and large unilamellar vesicles, we now propose that endophilin-LPD multivalent interactions might support negative Gaussian curvature at the neck region of membrane buds.

We believe, these new experiments raise both the physical and biologically relevant insight of our work. We address all other points raised by the reviewer below.

Comment 2

Other significant issues:

Why are the membrane condensates so different from solution? This would seem a highly relevant and interesting question that is largely ignored. Does the membrane inhibit condensate formation? If so, why?

ANS: The condensates formed in solution are driven by classical three-dimensional phase separation (Shin & Brangwynne, *Science*, 2017). The condensates formed on the membrane are a result of a quasi two-dimensional phase separation where proteins laterally diffusing on the membrane surface form a liquid like assembly (Banjade & Rosen, *eLife*, 2014). In the course of two-dimensional phase separation, the protein concentration in the solution remains significantly below (between 50 nM to 1 μ M in most cases) the bulk LLPS threshold and no three-dimensional structures form. Several past studies have shown two-dimensional protein condensates on membrane and our observations are not very different from those. A few examples are shown below:

Figure 1B from Case *et al. Science*, 2019;

Figure 1B from Su *et al. Science*, 2019;

Figure 5 from Zeng *et al. Cell*, 2018.

This is now clarified in the revised manuscript.

Changes in the manuscript:

We added the following text to the Results section (Page 22, Para 2)

“The membrane clusters formed well below the phase boundary of endophilin/TIL and endophilin/LPD systems (**Figure 3F-H**) in the bulk. The clusters resemble membrane condensates earlier observed with signaling protein complexes¹⁸ and postsynaptic density proteins⁶⁴. FRAP studies on the clusters showed partial photorecovery of TIL, PRM7, and LPD₈₅₀₋₁₂₅₀, indicating liquid-like properties of the two-dimensional clusters (*Supplementary figure 10B*).”

Comment 3

Why weren't these experiments conducted on GUVs to test if condensates bend the membrane or facilitate invagination?

ANS: This is a crucial point raised by both reviewer 1 and reviewer 3. As per the reviewer's suggestion, we performed experiments with reconstituted FEME components on giant unilamellar vesicles (GUVs) and large unilamellar vesicles (LUVs).

In the presence of reconstituted endophilin and LPD on GUV-membranes we did not observe any inner tubulation, suggesting that phase separation of these proteins do not generate

negative membrane mean curvature (such as inner tubulation), at least under our experimental conditions. Interestingly, we observed that binding of LPD to endophilin-covered membranes causes membrane-adhesion. This results into an increase in membrane tension, possibly by a mechanism involving adhesion-induced wrapping of tubules on the GUV surface. Membrane adhesion was also evident from TEM images with endophilin-covered LUVs in the presence of LPD. From these observations, we propose that membrane adhesion mediated by endophilin-LPD multivalent interactions could support negative Gaussian (i.e. saddle-shaped) curvature at the neck of the membrane buds formed at FEME sites. We share these insights in our revised manuscript.

Changes in the manuscript:

Added a new Results section (Page 28) entitled as, “Endophilin interacts with LPD on membranes to support membrane-membrane adhesion and budding necks”

Added new results to Figure 6, and new Supplementary Figures 14 and 15.

Comment 4

There are several speculations about the relevance of these results for regulation of FEME or endocytic protein assemblies, but these cannot go beyond speculations, because the relevance of these structures themselves has not been supported.

ANS:

We acknowledge that our work does not show direct correspondence between our in-vitro and cellular observations. That being said, Boucrot et al. (Ferreira *et al. Nat. Comm*, 2021; Boucrot *et al. Nature*, 2015) showed *in vivo* that transient endophilin-lamellipodin clusters form at the plasma membrane and that these can be stabilized through receptor engagement. In our revised manuscript, we have emphasized this shortcoming. We note that both referee 2 and 3 highlight the relevance of our mechanistic insight. Referee 2: “This is a very important manuscript as it constitutes the first in vitro reconstitution of the priming and initiation steps of FEME and it recapitulates the steps observed in cells in a controlled environment.” Referee 3: “Previously, the molecular mechanism was unknown as to how TIL and endophilin interact to facilitate FEME. This paper suggests that LLPS could play an important role in FEME, with endophilin clusters acting as priming sites for FEME through the recruitment of GPCRs and other endocytic proteins.”

Changes in the manuscript:

We now discuss the limitations of our model in the *Discussion* section and refrain from any speculations that are model-specific and might need additional experiments using the full-length proteins within the cellular system. Specifically, the modified discussion section (Page 33, para 3) reads as follows:

“Within the limitations of our current *in vitro* model, we propose the following mechanisms of carrier formation during FEME. On one hand, membrane remodeling can occur through enhancement of local endophilin N-BAR activity within the clusters and promotion of membrane scaffolding by rigid N-BAR assembly. On the other hand, TIL mediated enhancement in endophilin-rich clusters would also allow partitioning of various other proteins, including N-BAR protein BIN1, which recruits downstream protein machineries such involving Dynein to the FEME carriers⁵⁰. Curvature generation could therefore be facilitated by a local enrichment of BAR-proteins at the FEME sites. To understand how and when various other proteins, such as actin regulatory machineries, are engaged to facilitate carrier formation process would require a more rigorous model, ideally using reconstituted full-length proteins. In addition, how local composition of membrane phospholipids affect these interactions is a key question yet to be addressed.

Nevertheless, our study shows many of these interactions could now be understood in the light of protein-protein phase separation driven by multivalent interactions.”

Comment 5

For the phase diagrams, it is difficult to get a sense of the variance in these experiments. Are different experiments (and especially different preparations) as tight as the phase diagrams would appear to suggest? Put another way, what is a meaningful difference in these experiments?

ANS: We thank the reviewer for this important comment. We now show the variances in the phase boundaries for Figure 1F, and Figure 2E where comparative studies that used the differences in phase boundaries to compare protein-protein interactions are shown. These experiments are repeated at least two times (using independent samples) and only minor variations in the threshold concentrations were observed for some cases. The figures now include such variations.

Changes in the manuscript:

We revised figure 1E and figure 2F by including variations in the phase boundary wherever such were observed. We added to the results section (Page 9, para 2):

“The observed LLPS threshold concentrations for various endophilin mutants mostly remained unchanged between three trials performed using different sample preparations, except for the N-BAR domain that showed a small (of ~ 1 μ M) variation. The fact that the BAR-domain-only mutants (both N-BAR and Δ H0-BAR) showed LLPS at 2-fold lower concentration than the full-length endophilin (**Figure 1F**) indicated that either the disordered linker or the SH3 domain might suppress droplet formation in the full-length protein.”

Comment 6

And how would one know whether such differences are meaningful in a biological (rather than statistical) sense?

ANS: We agree with the referee that an ultimate verification that these differences are biologically meaningful would require careful quantitative work in cells (also see our response to *comment 4* above). As we mentioned under comment 4, reviewers 2 and 3 agree that our in-vitro work provides valuable insight, and there is recent precedent of related *in vitro* work being published in interdisciplinary journals (see e.g. Huang *et al. Science* 2019, Ghosh *et al. PNAS*, 2019). As we already mentioned in the same comment above, cellular context is provided through the published work of Boucrot *et al.* (Ferreira *et al. Nat. Comm*, 2021; Boucrot *et al. Nature*, 2015).

Comment 7

The notion that the intrinsically disordered linker of endophilin inhibits phase separation is interesting and surprising, in light of the many examples of IDPs forming condensates. This could have been another aspect for new insights.

ANS: We thank the reviewer for encouraging us discuss this interesting observation. Phase separation by multivalent proteins can be explained using a ‘stickers-and-spacers’ model proposed by Pappu and co-workers (Choi, Holehouse, & Pappu, *Annu. Rev. Biophys.*, 2020). The intrinsically disordered regions (IDRs) can favor phase separation by allowing conformational flexibilities to form three-dimensional networks by sticker-sticker interactions. However, IDRs are not necessarily the drivers of phase separation (Choi, Holehouse, & Pappu, *Annu. Rev. Biophys.*, 2020; Mittag, *Current Opinion*, 2021). Indeed, it has been shown that IDRs can inhibit LLPS as

well (Riback, Drummond, *Cell*, 2017). This opens new possibilities for studying the mechanistic details of phase separation behavior of BAR-domains with experimental biophysical techniques and computational modeling. We emphasized this point in the manuscript.

Changes in the manuscript:

We added the following text to the Results section (Page 9, para 2)

“The notion of disordered linkers playing inhibitory roles in BAR-proteins is interesting but not necessarily surprising. While intrinsically disordered regions (IDRs) can favor phase separation in many proteins by allowing conformational flexibilities to form three-dimensional networks they are not always the drivers of phase separation^{42, 43}. Indeed, it has been shown that IDRs can inhibit LLPS as well⁴⁴.”

Comment 8

And another: what differences between amphiphysin and FBP17 (both BAR-containing proteins) lead to one forming solid-like shell and suppresses LLPS while the other does not?

ANS: The differences between the effects shown by amphiphysin 1 and FBP17 can possibly be explained by the differences in their disordered linker length. The amphiphysin linker (382 aa) is 100 amino acid longer than FBP17 linker (282 aa). Therefore, the conformational entropy penalty upon partitioning into the droplet phase is expected to be larger in the case of Amph1 compared to FBP17. This would result into a higher tendency for Amph1 to remain at the droplet-aqueous phase interface. This may give rise to future studies into the linker behavior of BAR-proteins using computational modeling. We thank the reviewer for asking this question and have added this discussion to the manuscript.

Changes in the manuscript:

We added the following to the results section (Page 17, para 2):

“Interestingly, FBP17 did not exhibit similar surfactant-like behavior. A plausible mechanism would be that due to its longer linker length (100 amino acid longer than FBP17), the change in conformational entropy from aqueous phase to droplet phase⁵⁹ would be more negative for Amph1 than FBP17. Therefore, to minimize the entropic penalty upon droplet partitioning, Amph1 prefers to remain in the interfacial region of a droplet.”

Comment 9

LLPS is clearly an inappropriate description here because of how non-liquid some of the behaviors appear.

ANS: The signature liquid-like behavior of the condensates are their abilities to form spherical shapes that can coalesce into larger spherical assemblies, the mobility of the components inside condensates and their exchangeability with the dilute phase, and finally their ‘wetting’ behavior when in contact with a surface (Shin & Brangwynne, *Science*, 2017). We observed that droplets formed by endophilin, via both self-association and multivalent interactions, exhibit all these liquid-like behaviors. The comparatively slow photobleaching recovery of endophilin within the droplets could be a result of strong associative interactions between the protein molecules that slow down the molecular mobility of the protein components. It is not uncommon to observe a continuum between viscous liquid properties and visco-elastic gel like properties of biomolecular condensates (see e.g. Kaur et al. and Bannerjee, *Biomolecules*, 2019).

Specific proteins such as α -synuclein and tau have been characterized to undergo a solid-like transition that is facilitated by LLPS (Ray *et al. Nature Chemistry*, 2020; Wegmann *et al. EMBO*

J., 2018). Such transitions are associated with a conformation change within the droplets over the course of days. No such conformation changes or solid-like transitions were observed in our endophilin droplets (*Supplementary Figure 2* (CD data), *3C* (STED) and *3D* (TEM images).

Changes in the manuscript:

To clarify this aspect, we added the following to the Results section:

Page 4, para 2:

“Formation of liquid droplets was confirmed by their spherical appearances under transmitted light microscopy, their behavior to undergo coalescence, and their ‘wetting’ behavior on glass surfaces ²³ (**Figure 1A**, *Supplementary Figure 1A, 1B*).”

Page 8, para 1:

“Finally, characterization of the endophilin droplets with two additional alternative techniques, stimulated emission depletion microscopy (STED), and negative stain transmission electron microscopy (TEM) (*Supplementary Figure 3C, 3D*) did not show any inhomogeneous staining or solid-like structures inside the droplets as characterized in the case of disordered proteins such as synuclein or tau ^{32, 39}.”

Comment 10

LPD appears to be a major regulator of endophilin and FEME. It would seem valuable to extend the studies to full-length LPD rather than PRM7.

ANS: We thank the reviewer for this very helpful suggestion that helped us present our work in a more impactful way. This encouraged us to extend our study with longer LPD variants. The full-length LPD is 1250 amino acid long and cannot be purified. Hansen and Mullins (*eLife*, 2015) showed that a 400 amino acid long LPD variant (LPD⁸⁵⁰⁻¹²⁵⁰) that contains at least 4 endophilin-SH3 binding sites, can be purified as a recombinant protein using *E. coli*. We successfully purified the LPD⁸⁵⁰⁻¹²⁵⁰ in our lab and studied its phase behavior in bulk as well as on the membrane. The data for LPD⁸⁵⁰⁻¹²⁵⁰ have been included in Figures 2-6 of our revised manuscript.

Changes in the manuscript:

- i) We have added new data obtained with LPD⁸⁵⁰⁻¹²⁵⁰ in Figures 2-6.
- ii) We have complemented our Results sections with the observation with LPD⁸⁵⁰⁻¹²⁵⁰.

Reviewer #2 (Remarks to the Author):

Mondal, Baumgart and colleagues are reporting that protein-protein phase separation driven by multivalent interactions plays a key role in the initial steps of Fast Endophilin-mediated endocytosis (FEME) carrier formation.

FEME is a portal of entry into cells that is not always active (unlike for example, Clathrin-mediated endocytosis that is constitutive). FEME regulates key physiological functions (e.g. cell migration, rapid receptor uptake, cell signalling modulation), and its inhibition was recently shown to boost anti-cancer treatments using antibody-dependent cellular toxicity (ADCC)-mediating therapeutic antibodies (Chew HY et al. Cell 2020).

They provide evidence that i) the BAR domain of Endophilin drives liquid-liquid phase separation in crowded environment, ii) Endophilin binding partners (cargo receptor tail (bet1-Adenergic

receptor third intracellular loop (beta1-AR TIL) and accessory proteins Lamellipodin proline-rich motifs (Lpd PRM), FBP17 and Amphiphysin 1) partition into LLPS droplets, and iii) that such client proteins regulate LLPS formation depending on their molecular features. Then, the authors demonstrated that, iv) Endophilin undergoes LLPS upon binding to multiple PRM containing ligands even in the absence of crowding agents and, v) causes clustering of membrane bound beta1-AR TIL and Lpd PRM by two-dimensional phase separation. Interestingly, they report that Lpd PRMs pre-enrich Endophilin on membrane but only the addition of TIL triggers robust cluster formation.

This is a very important manuscript as it constitutes the first in vitro reconstitution of the priming and initiation steps of FEME and it recapitulate the steps observed in cells in a controlled environment. And importantly, it reveals the molecular mechanism by which FEME carriers are initiated and demonstrates how multivalent interactions between the key initial proteins drive they local concentration up to a phase transition into liquid-like condensates. This manuscript is an important addition to our understanding of how FEME works and complement the existing literature.

The findings may well prove to apply to other endocytic and intracellular membrane trafficking steps.

The experiments were carefully designed (with appropriated controls), executed and analysed and constituted a high-quality piece of work.

Our response

We are grateful to the reviewer for the positive feedback. Furthermore, the suggestions have been very useful to present our work in a more impactful way. Below, we address the points raised by the reviewer.

Comment 11

My only comments are the following:

The authors report the very interesting finding that Amphiphysin 1 does not form LLPS droplets (Sup Fig. 5) on its own and is recruited into a solid-like shell around Endophilin driven LLPS droplets (Figure 2F-H).

However, Amphiphysin 1 is tissue-restricted and mostly (if not only) expressed in the brain, whereas FEME was also observed in a variety of tissue-culture as well as primary cells from various non-brain organs (kidney, retina, skin and umbilical cord veins). Instead, Amphiphysin 2 (aka Bin1) is ubiquitously expressed, although the expression pattern of its isoforms vary.

Relevant to this work, Ferreira APA et al Nat Comm 2021 recently reported that the ubiquitously expressed isoform 9 of Amphiphysin 2 (Bin1 iso9) functions in FEME by recruiting Dynein onto FEME carriers. The cellular depletion of all Amphiphysin 1 and 2 isoforms did lead to a dysregulation of Endophilin spots, which was attributed to the failure to recruit Dynein (Ferreira APA et al Nat Comm 2021) but the hypothesis from Mondal and colleagues (the present manuscript) that it would also be due to the loss of the regulation of the size of the priming patch by the absence of Amphiphysin could also contribute to the phenotype.

Importantly, the two main Amphiphysin 2 isoforms (iso 9 and 10) that are ubiquitously expressed have much shorter linker between their BAR and SH3 domains and thus resemble much more to Endophilin than to Amphiphysin 1.

Whereas both Amphiphysin 1 and the long isoforms of Amphiphysin 2 bind to Clathrin and AP-2 (the motifs are in the exons skipped in the short isoforms 9 and 10) and function in CME.

Thus, it would be important to complement Figure 2 and Supplementary Figure 5 with one of these short ubiquitously expressed Amphiphysin 2 isoform (aka Bin1 iso 9 or 10) to validate the authors' conclusion that Amphiphysin forms a solid-like shell and antagonise LLPS of Endophilin because of its long disordered linker.

Such experiment is worth doing as both outcomes would be informative:

Should the short Amphiphysin 2 isoforms be recruited into LLPS and Amphiphysin 1 (and thus likely the long Amphiphysin 2 isoforms) are excluded, it would confirm the authors' hypothesis that the longer linkers drive the exclusion and an argument could be made for the size control by the long Amphiphysin isoforms.

Should the short Amphiphysin 2 isoforms be also restricted to the periphery of the LLPS, it would provide a basis for the proposition that the exclusion of Amphiphysin 2 (Bin1) to the rim of the forming FEME priming patches might help the recruitment of Dynein to the right location (the periphery of the patches) to apply the pulling tension and trigger the swift FEME carrier formation.

ANS: We appreciate reviewer's thoughtful suggestions. To test the suggested hypotheses, we purified BIN1 (iso 9). Interestingly, the partitioning of BIN1 into endophilin droplets was found to be homogeneous. Furthermore, BIN1, unlike Amphiphysin1 (Amph1), did not reduce the droplet size. This supports our hypothesis that the longer linker of Amph1 is the key molecular determinant that drives Amph1's peripheral distribution and droplet size regulation behavior.

In addition, we found BIN1, like endophilin, undergoes LLPS under crowding conditions. We examined BIN1 as a client for endophilin droplets and added the relevant data in Figure 2 and Supplementary figure 5C, 5D.

Changes in the manuscript

Added BIN1 data to Figure 2 in the main text and *Supplementary Figure 5C, 5D, and 8C.*

Notable changes in the main text:

Page 13, para 1:

“In addition to endophilin, various other BAR-domain proteins participate in FEME. Two additional N-BAR family proteins, amphiphysin 1 (Amph1) and BIN1, have been found in endophilin rich spots both at the leading edge of the plasma membrane and in the majority of FEME carriers formed upon receptor engagement in BSC-1 cells^{49, 50}. Amph1 was also reported to interact with the SH3 domain of endophilin *in vitro* via PRMs within its large disordered linker region, and this interaction has been implicated in CME of synaptic vesicles.⁵¹
⁵² The F-BAR family protein FBP17 is involved in regulating recruitment of LPD at the leading edge of cells during FEME and is also shown to co-localize with endophilin at sites of CME^{49, 53}. We then tested the partitioning of Amph1, BIN1 (the ubiquitously expressed isoform 9), or FBP17 into endophilin droplets. All three proteins strongly partitioned into the droplets, with K_{app} values of 352 ± 80 , 267 ± 20 , and 1130 ± 350 , respectively (**Figure 2C, Supplementary table 1**). Interestingly, Amph1 showed

an anisotropic partitioning behavior by accumulating preferentially at the droplet periphery compared to the droplet interior (**Figure 2B**).”

Page 17, para 2:

“The observed amphiphilic, surfactant-like properties suggest a potential role of Amph1 as a size regulator of endocytic protein assemblies in FEME. Amph1 is mostly expressed in the brain whereas BIN1 is more ubiquitously expressed and plays an important role in FEME by recruiting Dynein⁵⁰. We have already shown that the isoform 9 of BIN1, which has a short linker, similar to endophilin, also undergoes LLPS in the presence of PEG. Unlike Amph1, BIN1 (iso 9) did not show peripheral distribution when partitioned into endophilin droplets (**Figure 2F, 2G**). In addition, we did not observe droplet size regulatory behavior of BIN1 at the concentration range (0.05-1 μ M) where Amph1 caused significant reduction in droplet size (**Figure 2H**, and *Supplementary Figure 8C*). These data strongly suggest that surfactant-like behavior of Amph1 is driven by its long, disordered linker.”

Comment 12

The findings may well prove to apply to other endocytic and intracellular membrane trafficking steps, but the current data does not support the broad title: “[...] involved in Clathrin-independent endocytosis [...]” (lines 1-2), as it solely focused on FEME.

Clathrin-mediated endocytosis is broader than FEME and comprises processes as mechanistically different as the micropinocytosis, CLIC/GEEC, Massive endocytosis (MEND), EGFR non-Clathrin endocytosis (EGFR-NCE) and pathways operating according to the GL-Lect hypothesis, for which the mechanisms here may or may not apply.

I would not think that a title restricted to “[...] involved in Fast Endophilin-Mediated Endocytosis [...]” would decrease in any way the importance of this study. As mentioned above, FEME is involved in important physiological processes and its manipulation may have useful therapeutic applications.

ANS: We agree with this comment and have changed the title.

Changes in the manuscript

The new manuscript title is: “Multivalent Interactions between Molecular Components Involved in Fast Endophilin Mediated Endocytosis Drive Protein Phase Separation”

Comment 13

The text reports a “significant drop in the number of large droplets” (line 336) but statistics are missing on the corresponding panel (Supplementary Fig. 8B)

ANS: We agree with this comment: the significance analysis to the data has been added.

Changes in the manuscript

Supplementary figure 8B has been replaced with a new figure (*Supplementary figure 8C*) that contains the estimated ‘effect size’ in terms of Cohen’s *d* values.

Reviewer #3 (Remarks to the Author):

Several recent studies have identified roles of liquid-liquid phase separation (LLPS) in endocytic processes. In this work, Mondal et al examine molecular interactions involved in Fast Endophilin

Mediated Endocytosis (FEME), a clathrin-independent endocytic pathway. The authors demonstrate that endophilin, a key player in FEME, can undergo LLPS in vitro in the presence of crowding agents. Multivalent binding partners of endophilin that are involved in FEME – including the third intracellular loop (TIL) of the beta-1-adrenergic receptor (a G-protein coupled receptor, GPCR), as well as the proline-rich motifs (PRM) of lamellipodin (LPD) – can trigger LLPS (in the absence of crowding agents) in bulk solutions and on membranes. On membranes, TIL has a higher propensity to phase separate with endophilin, compared to PRM7.

Previously, the molecular mechanism was unknown as to how TIL and endophilin interact to facilitate FEME. This paper suggests that LLPS could play an important role in FEME, with endophilin clusters acting as priming sites for FEME through the recruitment of GPCRs and other endocytic proteins. These clusters could enhance protein activity at the membrane and lead to formation of FEME transport carriers. This work will contribute to the emerging role of LLPS in endocytosis and will be of interest to the membrane biophysics and LLPS communities. I do, however, have several major and minor suggestions for revision:

Our response:

We thank the reviewer for their supportive feedback to our project and for the careful evaluation of the manuscript. The suggestions given helped us in establishing our findings on even stronger grounds. We have carefully addressed all the comments by the reviewer below.

Comment 14

Major:

1. The authors' interpretation of the FRAP data (Fig. 1) is that endophilin-rich droplets have a biphasic state, with a mobile phase and immobile phase. Are there other biophysical measurements that could back up this claim?

ANS: We appreciate the reviewer raising this important point and suggesting to test our hypothesis of a co-existing biphasic state by characterizing the droplet material properties with alternative methods. We have now performed negative-stain transmission electron microscopy (TEM) and super-resolution optical microscopy (STED) analysis of the endophilin droplets. Neither of these techniques showed that endophilin droplets form solid-like fibrillar structures, as observed in the case of α -synuclein (Ray *et al. Nature Chemistry*, 2020) or tau (Wegmann *et al. EMBO J.*, 2018 37:e98049). We therefore modify our interpretation from the FRAP studies and use the TEM data to support our alternate hypothesis that the droplets undergo a gel-like transition via protein-protein interactions.

Changes in the manuscript:

Modified the concluding sentence Page 5, para 2 to:

“Interestingly, the average halftime ($t_{1/2}$) of photorecovery obtained from the exponential fits did not vary significantly with different PEG concentrations, indicating that the diffusion properties of the droplets within our time-scale of observation are dominated by the fast-diffusing component (*Supplementary Figure 1C*).”

Removed the previous hypothesis: “This observation suggests that endophilin-rich droplets transition into a biphasic state consisting of a mobile phase with unchanged diffusion properties, and a second, solid-like phase where the protein is immobilized.”

Included the STED and TEM images to the *Supplementary figure 3C, 3D* and discussed in the Results section (Page 8, para 1):

“Finally, characterization of the endophilin droplets with two additional alternative techniques, stimulated emission depletion microscopy (STED), and negative stain transmission electron microscopy (TEM) (*Supplementary Figure 3C, 3D*) did not show any inhomogeneous staining or solid-like structures inside the droplets as characterized in the case of disordered proteins such as synuclein or tau^{32, 39.}”

Comment 15

The authors did a good job describing the importance of the BAR domain to LLPS of endophilin. I do wonder about one variant that was not tested: What is the phase behavior of just the BAR domain, with both H0 and SH3 deleted?

ANS: As per the reviewer’s suggestion we purified the BAR domain (without H0, linker, and SH3 domain) and found that the phase behavior remained unchanged compared to the N-BAR domain. The data is added to our Figure 1F. This further establishes our hypothesis that endophilin LLPS is driven entirely by its BAR-domain on a stronger ground. We thank the reviewer for this helpful suggestion.

Changes in the manuscript:

Included the Δ H0-BAR domain data to figure 1F and added the following text in Results (Page 8, para 2):

“Deletion of the H0 region from the N-BAR domain did not cause any significant change to its threshold concentration for phase separation (**Figure 1F**).”

Comment 16

Can the authors further comment on whether PRM7 is a good model for the C-terminal region of LPD, and why they chose this model? The C-terminal domain of LPD is much longer and contains 10 PRMs.

ANS: We understand the importance of showing a validation for our PRM7 model using the natural LPD-C terminal region. We therefore purified a 400 amino acid long disordered segment (aa 850-1250) of LPD-C terminus that contains at least 4 endophilin-SH3 binding domains. Our experiments demonstrate that, similar to our PRM7 model, ^{LPD850-1250} drives liquid-like droplet formation in bulk in the presence of endophilin, forms clusters on supported bilayers in the presence of endophilin, and partitions into endophilin droplets formed in the presence of PEG. The only observed differences are - i) phase separation occurs in a different protein concentration regime of the phase diagram, and ii) partition co-efficient (K_{app}) for LPD is 20 times higher than that of PRM7.

These data suggest that PRM7 offers a qualitative model for LPD that conserves the nature of the interactions with endophilin whereas the strength of specific interactions can be different. Therefore, observations such as relative clustering tendencies between TIL and LPD, that depend on the strength of specific interactions, might appear different between a PRM7 model, shorter fragments such as LPD₈₅₀₋₁₂₅₀, and the full-length protein. To be consistent with current observations, we refrain from drawing conclusions such as TIL showing stronger clustering abilities on the membrane since those were derived from our earlier observations using these model proteins.

Changes in the manuscript

LPD₈₅₀₋₁₂₅₀ data has been included in Figures 2-6 and corresponding Results sections.

Notable change in the Results sections:

Page 11, para 1:

“The second client we considered was the C-terminal domain of LPD. The entire C-terminal domain of LPD is a 658 amino acid long (aa 593-1250) disordered sequence that contains 10 PRMs, each 12-13 amino acids long. These PRMs have been shown to bind the endophilin SH3 domain¹⁶. To facilitate purification, we worked with a relatively shorter (400 aa) fragment of the LPD C-terminal domain (850-1250, LPD⁸⁵⁰⁻¹²⁵⁰,⁴⁸. LPD₈₅₀₋₁₂₅₀ contains 4 out of 10 PRMs and is thus expected to exhibit multivalent interactions with the endophilin-SH3 domain. In order to be able to vary that PRM domains within LPD⁸⁵⁰⁻¹²⁵⁰ dominate its interactions with endophilin, we designed a synthetic mimic of the LPD C-terminal domain that consists of a heptameric repeat of a single PRM of LPD (aa 970-981) connected via flexible (Gly-Gly-Ser)₄ linkers (PRM7).”

Page 22, Para 2:

“The fact that we observe similar behavior comparing the simple PRM7 peptide (multiple repeats of a single PRM separated by oligo-GGS spacers) and the more complex LPD⁸⁵⁰⁻¹²⁵⁰ suggests that the behavior of the latter is dominated by its PRMs.”

Comment 17

Fig. 2F, G are very interesting and indeed suggest that amphiphysin localizes at the interface of endophilin condensates, like a surfactant. However, is the size data in Fig. 2H due to amphiphysin (Amph) acting as a surfactant that stabilizes smaller droplets, or is Amph inhibiting LLPS? In other words, is the total volume of the condensed phase approximately conserved with vs. without Amph?

ANS: We appreciate the reviewer for asking this question and thank them for this excellent suggestion of estimating the total volume of the condensed phase. We determined this in an indirect way, by estimating the protein concentration in the dilute phase. After the droplet formation at various amphiphysin (Amph1) concentrations, we centrifuged the droplets and estimated the remaining protein concentration in the dilute phase via Bradford assay. If there is an inhibition of LLPS by amphiphysin, that would increase the protein concentration in the dilute phase. However, if Amph1 acted as a surfactant only, we expect to not observe a change in the protein concentrations in the dilute phase.

In our experiment, no changes in the estimated protein concentration in the dilute phase were observed in the presence or absence of Amph1. In order to be accurate in estimating the endophilin concentrations in the dilute phase, we generated a calibration curve using known concentrations of endophilin itself and made sure that the estimated protein concentrations in the dilute phase remains within the calibration range. We note that addition of 0.05-1 μ M of Amph1 would only have a minor effect (< 10%) on the estimated protein concentration. Our estimation showed a dilute phase concentration of 10 ± 1 μ M, which is similar to the threshold protein concentration for LLPS in the presence of 10% PEG as expected from our endophilin-PEG boundary. Observing no changes in the dilute phase protein concentrations upon addition of amphiphysin in the range of 0.05 – 1 μ M, supports our hypothesis that Amph1 stabilized smaller droplets by acting like a surfactant. We include the following data and discuss this observation in our manuscript.

Changes in the manuscript:

Added to the Results (Page 16, paragraph 2):

“We asked if a reduction in the droplet size was correlated with a change of the protein volume fractions in the dilute and the condensed phase. To indirectly estimate the protein volume fractions in the dilute and the condensed phases, we separated the droplets formed by centrifugation and estimated the protein concentrations in the dilute phase by Bradford assay. Inhibition of LLPS would cause a reduction in the protein volume fraction in the condensed phase that would be reflected by an increase in the estimated protein concentration in the dilute phase. The estimated protein concentration did not show significant changes in the absence or in the presence of 0.05-1 μM of Amph1 (*Supplementary Figure 8D*). These data suggest that the droplet size reductions by amphiphysin (up to 1 μM) can be attributed to a surfactant-like behavior, as opposed to Amph1 inhibiting homotypic LLPS of endophilin.”

Comment 18

In Fig. 3C, D: It would have been preferable if the authors used the same fluorophore to label endophilin. Also, the recoveries of PRM and TIL are so fast relative to the image acquisition frequency that that it's unclear the fits are quantitatively accurate. It would be helpful if the authors can comment on these issues.

ANS: We thank the reviewer for raising these points. To answer the first part of the question, we avoided using the fluorophore combinations Alexa 594/Alexa 633 or Alexa 594/Alexa 647 because of their strong spectral overlap, that would potentially influence 2-color imaging. Spectral overlap is weaker between Alexa 488/Alexa 594, Alexa 488/Alexa 633, and Alexa488/Alexa 647 fluorophore pairs and hence these fluorophore combinations were used for our 2-color FRAP studies.

We acknowledge the reviewer's concern that fluorescent tags can alter protein mobility (Zanetti-Domingues, *et al. PloS one*, 2013) and therefore it is ideal to use the same fluorophore on a single protein, particularly when comparing the protein mobility in two different sets of experiments. However, in this study our main goal was to demonstrate the significant differences between the mobilities of endophilin and its multivalent binding partner (TIL or PRM7 or LPD⁸⁵⁰⁻¹²⁵⁰) rather than comparing the mobilities of endophilin itself in two different kinds of droplets (Endo/TIL vs Endo/LPD). We clarify this in our revised version of the manuscript.

Answer to the second part of the question:

We adjusted the image acquisition frequencies of our confocal microscope according to the measured recovery rates (e.g. 3 s per frame for TIL/endo whereas 0.17 s per frame for PRM7/endo) so that we collect data points within the rise time. During exponential fitting, the lower bound of the time-constant was set to the time resolution of our image acquisition so that the fitting would not result into a value lower than what we could measure. Therefore, we believe our rate estimations are accurate within the error limit shown. We now discuss this in the *Methods* section of our manuscript.

Changes in the manuscript

Added to the Results section (Page 18, para 1):

“In order to minimize spectral overlap between fluorophores affecting the fluorescence recovery profiles, endophilin labeled with either Alexa 594 (while using TIL-Alex 488; **Figure 3C**) or Alexa 488 (while using PRM7-Alexa 633 or LPD⁸⁵⁰⁻¹²⁵⁰-Alexa 647; **Figure 3D, 3E**) were used. TIL, PRM7, and LPD⁸⁵⁰⁻¹²⁵⁰ showed

rapid photorecovery and a greater extent of recovery compared to endophilin (**Figure 3C-E**). The mobility of endophilin itself in three different types of droplets could not be compared since different fluorescent tags were used to label endophilin in these cases. However, the reduced fractional mobility shown by endophilin in all three types of droplets is indicative of the formation of a gel-like state via BAR domain-driven self-association that we had observed in the case of droplets formed in the presence of PEG (**Figure 1**).”

Added to the Methods section (Page 40, para 3):

“The image acquisition frequencies were adjusted according to the measured recovery rates (e.g. 3 s per frame for TIL/endophilin droplets whereas 0.17 s per frame for PRM7/endophilin droplets) so that sufficient data points are collected within the rise time of the recovery profile. During exponential fitting, the lower bound of the time-constant was set to the time resolution of our image acquisition so that the fitting would not result in a value lower than what we could measure.”

Comment 19

It is important to note what buffer conditions (in addition to PEG) were used for these experiments, since ionic strength, pH, etc. are important regulators of LLPS. Although such information is provided in the Methods, the buffers used in the experiments should also be clearly stated in the captions. For example, this seems quite important for Fig. 3.

ANS: Our figure captions are now updated with the buffer conditions. We thank the reviewer for pointing out this oversight.

Changes in the figure captions

Included the buffer conditions in all figure captions.

Comment 20

How do the authors reconcile what appears to be different trends between Fig. 5 (TIL has stronger clustering than PRM7 on the membrane) vs. Fig. 3 (endophilin-PRM system requires lower concentrations for LLPS than endophilin-TIL)?

ANS: This is a very important point, we thank the reviewer for bringing this up. We believe that the observed clustering of tethered His-tagged proteins on supported bilayer can be influenced by their lateral mobility on the bilayer. Compared to TIL (Alexa 488 labeled), a reduced mobility on bilayer was shown by PRM7 (633 labeled) ($t_{1/2}$ 18±1 s, 56% recovery as opposed to $t_{1/2}$ 6.8±0.3 s, 94% recovery for TIL-Alexa 488) as observed in our FRAP studies (*Supplementary figure 10A*). This restricted mobility of PRM7 could be a result of potential membrane-interactions of the conjugated fluorophore, Alexa 633, that has been found to show a stronger membrane interaction compared to other commercial fluorophores used in this study: Alexa 488, Alexa 594, and Alexa 647 (Hughes *et al.*, *Plos One*, 2014). The restricted mobility might affect the rate of cluster formation and the extent of clustering. This hypothesis is supported by our recent observation with Alexa 647 labeled LPD⁸⁵⁰⁻¹²⁵⁰ on bilayer. LPD⁸⁵⁰⁻¹²⁵⁰-Alexa 647 shows a similar recovery profile ($t_{1/2}$ 7.6±0.1 s, 92% recovery) as TIL and its clustering tendency in the presence of endophilin is comparable to that of TIL. We added this discussion to our manuscript as a possible explanation for the observed different trends of clustering between PRM7 compared to TIL and LPD⁸⁵⁰⁻¹²⁵⁰.

Changes in the manuscript

Included the following text to the Results section (Page 25, para 1):

“TIL showed a comparatively greater extent of clustering ($R = 153$ nm) than PRM7 ($R = 69$ nm) (Figure 5A-C, Supplementary table 3). This difference could be due to stronger membrane affinity of Alexa 633-labeled PRM7 than Alexa 488-labeled TIL on supported bilayer⁶⁰ that would slow down its assembly into clusters.”

Comment 21

The authors note in the discussion that their lab is currently working to understand how endophilin-TIL interactions ultimately affect membrane curvature and trafficking. Do the authors have any data suitable for sharing at this time to address these tantalizing questions?

ANS: We thank the reviewer for asking about this intriguing research direction. Indeed, we have carried out more experiments to understand the effect of multivalent interactions mediated by endophilin and its binding partners on membrane curvature. Our new observations suggest that multivalent binding partners of endophilin can facilitate membrane adhesions that could support the formation of membrane buds at FEME sites.

Specifically, we formulated three hypotheses and tested them in turn: 1) LPD enhances endophilin’s curvature generation capacity through local enrichment, 2) phase separation of LPD generates negative membrane curvature and 3) multivalent interactions between membrane-bound endophilin and endophilin-bound lamellipodin can stabilize negative Gaussian curvature at the necks of budding endocytic vesicles and tubules. Our observations from GUY imaging suggests that phase separation neither enhances the (positive mean-) curvature generation through endophilin nor generates negative membrane curvature. Instead we observed that binding of LPD to endophilin-covered membranes causes membrane-adhesion. This results into an increase in membrane tension, possibly by a mechanism involving adhesion-induced wrapping of tubules on GUY surface. Membrane adhesion was also evident from TEM images with endophilin bound GUYs in the presence of LPD. From these observations, we propose that membrane adhesion mediated by endophilin-LPD multivalent interactions could support negative Gaussian curvature at the neck of the membrane buds formed at FEME sites.

As per the reviewer’s request, now we share our initial observations with GUYs and LUYs in our revised manuscript.

Changes in the manuscript:

Added a new Results section (Page 28) entitled: “Endophilin interacts with LPD on membranes to support membrane-membrane adhesion and budding necks”

Added new results to Figure 6, and new Supplementary Figures 14 and 15.

Comment 22

Minor:

Line 38: Insert comma after “protein”

ANS: Done.

Comment 23

Line 54: Did the authors intend to write “membraneless organelles,” instead of “membrane organelles”?

ANS: We changed it to ‘membraneless organelles’.

Comment 24

Line 60: Provide the in-text citation to the Day et al paper (ref. 57).

ANS: We now add the in-text citation as requested.

Comment 25

Line 126: It would be helpful to provide a conclusion to this paragraph – I believe the authors are saying that the CD data suggests that PEG and LLPS do not cause protein conformation changes.

ANS: We have added now the following conclusive sentence (Page 7, para 1):

“Overall, the results from our CD experiments suggest that i) PEG itself does not induce a protein conformation change in endophilin, and ii) transitioning into the droplet phase does not induce a significant conformation change for endophilin.”

Comment 26

Supplementary Fig. 3B: It may be helpful to add statistical testing here.

ANS: Statistical significance has been added to this data.

Comment 27

Line 144: Comma should be changed to period.

ANS: Corrected this issue.

Comment 28

Figure 1E is missing an x-axis label.

ANS: Fixed this issue.

Comment 29

Lines 165-169: What data or figure is this text referring to? Is it Supplemental Fig 8A?

ANS: Amphiphysin N-BAR droplet images and its phase boundary in the presence of 10% PEG are now provided in *Supplementary figure 5B, 5D*. We thank the reviewer for pointing out this issue.

Comment 30

Given the importance of the radial average autocorrelation function in the analysis here, it would be useful to elaborate a bit further on this method (even briefly), beyond just citing reference 55.

ANS: In the Methods section we now elaborated upon the details of radial average correlation analyses.

REVIEWERS' COMMENTS

Reviewer #1 (Remarks to the Author):

The authors have made a commendable effort in addressing the concerns brought up by our originally critical review. Especially the observations of induced membrane curvature and phase separation induced by LPD add to the physiological relevance and plausibility of the original observations.

Our only remaining concern, which is relatively minor, is the non-liquid appearance of the condensates on membranes. The authors point out that sphericity and coalescence of the 3D droplets is important evidence to their liquid nature. On the other hand, the 2D clusters in Figs 4-6 appear irregular and small, rather unlike liquids. The immobile fractions measured by FRAP further suggest not-quite-liquid behaviors, already hinted at by the gel-like behaviors in 2D. It would be useful for future readers to know how consistent these specific aspects of the observations are (i.e. are the condensates sometimes disk-like? how repeatable is the FRAP behavior?) and what factors may yield the apparently distinct behaviors in 2D versus 3D.

Reviewer #3 (Remarks to the Author):

I have read the revised manuscript and the response letter from the authors, and I am pleased to recommend this manuscript for publication. The authors addressed my questions. Fig. 6 adds nicely to the impact of this work, while pointing to the need for further research. I do have one small note: On line 247, "very" should be changed to "verify."

We thank all the reviewers for carefully evaluating our revised manuscript. We address the reviewer comments below.

Reviewer #1 comments

Our only remaining concern, which is relatively minor, is the non-liquid appearance of the condensates on membranes. The authors point out that sphericity and coalescence of the 3D droplets is important evidence to their liquid nature. On the other hand, the 2D clusters in Figs 4-6 appear irregular and small, rather unlike liquids. The immobile fractions measured by FRAP further suggest not-quite-liquid behaviors, already hinted at by the gel-like behaviors in 2D. It would be useful for future readers to know how consistent these specific aspects of the observations are (i.e. are the condensates sometimes disk-like? how repeatable is the FRAP behavior?) and what factors may yield the apparently distinct behaviors in 2D versus 3D.

Our Response

We believe that the size and shape of the membrane clusters are indicative of the nucleation regime of phase separation which is also shown in various protein-clusters on membrane. We agree with the reviewer that partial photorecovery of the membrane condensates are indeed indicative of their gel-like behavior which was also observed in the case of 3D condensates. Similar gel-like behavior of both membrane and bulk condensates imply that this could be an endophilin-specific phenomena. These points are clarified in the manuscript now.

Changes in the manuscript: Included the following statements on page 22, para 2:

“Unlike the three-dimensional bulk condensates, the two-dimensional membrane condensates are irregularly shaped and smaller in size. Such cluster appearance is indicative of the nucleation regime of phase separation on two-dimensional surface ²⁴.”

“FRAP studies on the clusters showed partial photorecovery of TIL, PRM7, and LPD⁸⁵⁰⁻¹²⁵⁰ (*Supplementary figure 10B*), indicating that the two-dimensional membrane clusters exhibited partial liquid-like and partial gel-like behavior that was observed in the case of endophilin driven condensates formed in the bulk (Fig. 1c-e and Fig. 3c-e).”

Reviewer #2 comments

I do have one small note: On line 247, "very" should be changed to "verify."

Our Response:

We fixed this typo on line 222 (current line number). Thanks.